# LOW RANK QUANTIZATION ADAPTATION FOR LARGE LANGUAGE MODEL

## ABSTRACT

As the parameters of Large Language Models (LLMs) increase, quantization has emerged as a potent strategy for model compression and acceleration. Concurrently, Low-Rank Adaptation (LoRA) has been recognized as an effective method for enhancing LLM performance. However, integrating LoRA with quantization presents significant challenges, particularly in preserving the quantization format after model optimization. In this paper, we introduce Low rank Quantization Adaptation (LoQA) for LLM, a novel approach that effectively fine-tunes holistic quantization parameters. Specifically, we first propose Holistic Quantization Low-Rank Adaptation (HQ-LoRA), a new perspective on the quantization operator that is compatible with LoRA. This approach enables simultaneous fine-tuning of **all** parameters (scale and zero point), yielding notable improvements in model performance. Thanks to the expanded optimization landscape, LoQA is broadly applicable to various Post-Training Quantization (PTQ) techniques, ensuring better generalizability in practical deployments. To address the varying magnitudes of integer weights under different bit-widths, we further propose Quantized Bit-Aware Scaling (QBAS), a strategy that adjusts the LoRA scaling factor based on the current bit-width. This approach normalizes the influence of integer weights across different quantization levels, enhancing the efficiency and stability of the fine-tuning process. Compared to existing methods, LoQA consistently achieves performance gains across a wide range of models, proving its effectiveness and adaptability. Code is available in the supplementary materials.

## 1 INTRODUCTION

In recent years, large language models (Zhang et al., 2022; Le Scao et al., 2023; Touvron et al., 2023a;b; Bubeck et al., 2023) have demonstrated remarkable performance across various fields, attracting significant attention. However, the increasing number of parameters in these models has made training and fine-tuning progressively more challenging. This has led to a research focus on efficiently enhancing model performance on diverse tasks using massive datasets, thereby facilitating the deployment and utilization of LLMs by researchers and the general public.

Parameter-efficient fine-tuning (Xu et al., 2023; Hu et al., 2021; Kopiczko et al., 2023; Liu et al., 2024a) and quantization (Xiao et al., 2023; Lin et al., 2023; Frantar et al., 2022; Shao et al., 2023; Ma et al., 2024) have emerged as prominent methods for improving training efficiency and compressing models. Parameter-efficient fine-tuning techniques aim to minimize the number of fine-tuning parameters and computational complexity. These techniques enhance model performance while reducing fine-tuning costs, time, and computational resource consumption. For example, QLORA efficiently fine-tunes a 65B parameter model on a 48GB GPU using Low Rank Adapters and innovative 4-bit quantization (Dettmers et al., 2023). The low-rank adaptation (LoRA) (Hu et al., 2021) method reduces the number of fine-tuning parameters through low-rank matrix multiplications. This approach decreases memory usage during gradient updates and accelerates training speed. Additionally, freezing parameters in the backbone network during optimization allows for the integration of quantization methods. Mapping backbone network parameters to low-bit representations further improves training efficiency. A series of post-training quantization methods (Frantar et al., 2022; Xiao et al., 2023; Lin et al., 2023; Frantar et al., 2022; Shao et al., 2023; Ma et al., 2024) can quickly produce high-performance low-bit quantized models for the backbone network. The integration of quantization and parameter-efficient fine-tuning presents substantial challenges within neural net-

work optimization. Notably, maintaining the quantized format of the backbone network proves difficult following the integration of fine-tuned parameters. Initially, QLoRA (Dettmers et al., 2024) addresses this issue by employing post-training quantization to preserve the structure post-fusion. However, this method partially compromises the precision of fine-tuned parameters, impacting the overall accuracy of the model. To tackle this, QA-LoRA (Xu et al., 2023) constrains the dimensions of low-rank matrices, allowing the fine-tuning parameters to be incorporated directly into the zero points of the quantized backbone network. This ensures the stability of the quantization fixed points during parameter fusion, although it restricts the optimization space for fine-tuned parameters, thus capping potential performance gains for the language model.

In response, this paper introduces a novel approach named Low-Rank Quantization Adaptation (LoQA). This method enhances all quantized parameters with an efficient fine-tuning module through two key components: Holistic Quantization Low-Rank Adaptation (HQ-LoRA) and Quantized Bit-Aware Scaling (QBAS). HQ-LoRA provides a new perspective on the quantization operator, making it compatible with LoRA while maintaining mathematical equivalence to the original operator. Conceptually, if the quantization zero points in the backbone network are viewed as translational operations on intra-group weight parameters, the scale parameters then serve as scaling transformations that adapt these parameters to the quantization range. HQ-LoRA enables the simultaneous fine-tuning of all quantization parameters (scale and zero point), significantly expanding the optimization space. Concurrently, it preserves the quantized structure of the backbone network, ensuring that the quantization fixed points remain stable. To address the varying magnitudes of integer weights under different bit-widths, QBAS adjusts the LoRA scaling factor based on the current bit-width, normalizing the influence of integer weights across different quantization levels. This approach enhances the efficiency and stability of the fine-tuning process. LoQA comprehensively optimizes both sets of quantization parameters through gradient-based methods, thereby broadening the optimization space. The fine-tuning of the two sets of quantization parameters under this low-rank framework minimizes both time and computational expenses, yielding an optimized quantized model efficiently.

In summary, our contributions are as follows:

- **A novel perspective on quantization:** We introduce Holistic Quantization Low-Rank Adaptation (HQ-LoRA), which expands the optimization space for fine-tuning all quantized parameters. Through a comprehensive analysis of the dequantization process, HQ-LoRA efficiently fine-tunes all quantized parameters, significantly enhancing the model's capacity.

- **An innovative LoRA scaling strategy:** We propose the Quantized Bit-Aware Scaling (QBAS) technique, which dynamically adjusts the LoRA scaling factor based on the current bit-width. This approach normalizes the influence of integer weights across different quantization levels, thereby enhancing the efficiency and stability of the fine-tuning process. QBAS is particularly effective when dealing with varying magnitudes of integer weights under different bit-widths, ensuring consistent performance across diverse quantization settings.

- **Empirical validation of significant performance improvements:** Extensive experiments demonstrate that LoQA consistently outperforms previous fine-tuning methods that maintain quantized formats, and in many cases, matches the performance of state-of-the-art 4+16 bit methods. Notably, in ultra-low bit-width scenarios, LoQA's effectiveness is even more pronounced, with its 2-bit version surpassing the current 2+16-bit state-of-the-art method by 4.7% and even outperforming the original 16-bit model.

## 2 RELATED WORK

**Parameter-efficient fine-tuning (PEFT).** Parameter-efficient fine-tuning techniques aim to minimize the number of trainable parameters and computational complexity during model adaptation. For instance, methods like Low-Rank Adaptation (Hu et al., 2021) reduce the number of tunable parameters by learning low-rank matrices, which has proven to be an effective strategy for fine-tuning large language models. Recent research on Parameter-Efficient Fine-Tuning focuses on enhancing the performance of LoRA with the same parameter budget (Liu et al., 2024a), while proposing new

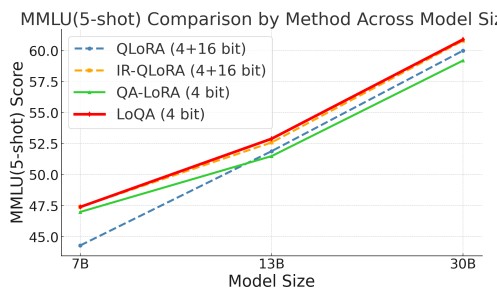 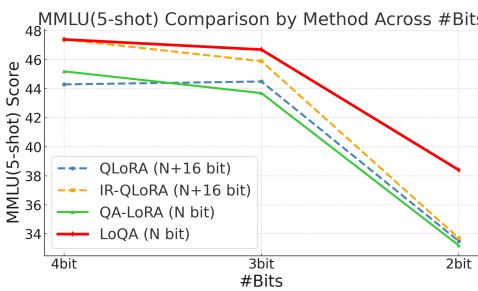

(a) Accuracy variation across model sizes.  (b) Accuracy variation with different bit-widths.

Figure 1: Performance analysis of LoQA across various configurations. (a) Demonstrates the scalability of LoQA across different model sizes. (b) Illustrates the robustness of LoQA under different quantization bit-widths. LoQA exhibits significant improvements over previous best methods that maintain the quantized format when combining LoRA and quantization methods. Notably, LoQA achieves performance comparable to state-of-the-art N+16 bit approaches that combine LoRA and quantization. These results underscore the efficacy and versatility of LoQA in enhancing model performance while maintaining low bit-width quantization.

fine-tuning methods that further reduce the number of tunable parameters while maintaining or improving efficiency (Ren et al., 2024; Gao et al., 2024; Azizi et al., 2024; Jiang et al., 2024; Meng et al., 2024; Kopiczko et al., 2023).

**Quantization of LLMs.** As LLMs scale up in parameter size, quantization has emerged as a powerful technique for model compression and acceleration, broadly classified into Post-Training Quantization (PTQ) and Quantization-Aware Training (QAT). PTQ is a key technology for speeding up and deploying LLMs. Recent work has focused on addressing outliers in both parameters and activations to improve the robustness and performance of quantized models (Xiao et al., 2023; Lin et al., 2023; Frantar et al., 2023; Shao et al., 2023; Ma et al., 2024; Ashkboos et al., 2024; Liu et al., 2024b). While QAT can enhance the performance of quantized models, its use in LLMs is limited due to the high cost of training. Vector quantization methods have also been introduced recently (Tseng et al., 2024; Egiazarian et al., 2024), which offer good precision but come with significant computational overhead. Our research mainly focuses on uniform quantization, which is more suitable for hardware implementation and offers faster inference speeds.

**Fine-Tuning of Quantized Parameters.** Techniques such as QLoRA(Dettmers et al., 2023), IR-QLoRA(Qin et al., 2024), LQ-LoRA(Guo et al., 2023), and LoftQ(Li et al., 2023) quantize model parameters into low-bit representations, followed by the addition of LoRA modules for fine-tuning. However, these approaches require the integration of floating-point LoRA modules with the quantized weights, leading to the restoration of model weights to floating-point format, preventing direct use of the quantized weights. In contrast, PEQA (Kim et al., 2024) employs a simple round-to-nearest (RTN) method for low-bit quantization and fine-tunes the quantized model's step size to adapt to downstream tasks, allowing the quantized model to be directly utilized post-fine-tuning. EfficientQAT (Chen et al., 2024) improves upon PEQA by replacing the simple RTN method to provide a better starting point for fine-tuning. The closest related method to ours is QA-LoRA (Xu et al., 2023), which redesigns the LoRA module to seamlessly integrate with zero-points. However, QA-LoRA requires zero-points to be in floating-point format, limiting its practical applicability. Additionally, it can only merge with zero-points, which constrains its overall performance, especially when fine-tuning on large downstream datasets.

## 3 LOW-RANK QUANTIZATION ADAPTATION

This section introduces LoQA, a novel two-stage quantization and fine-tuning approach designed to achieve high-performance quantized models for downstream tasks under resource constraints. The first stage employs a limited amount of calibration data to perform efficient post-training quantization (PTQ), yielding initial quantized weights $W^{\text{Int}}$ and quantization parameters (step sizes $S$ and zero points $Z$). This approach enables fine-tuning of the actual quantized model during the second stage, significantly reducing memory requirements. In the second stage of fine-tuning for

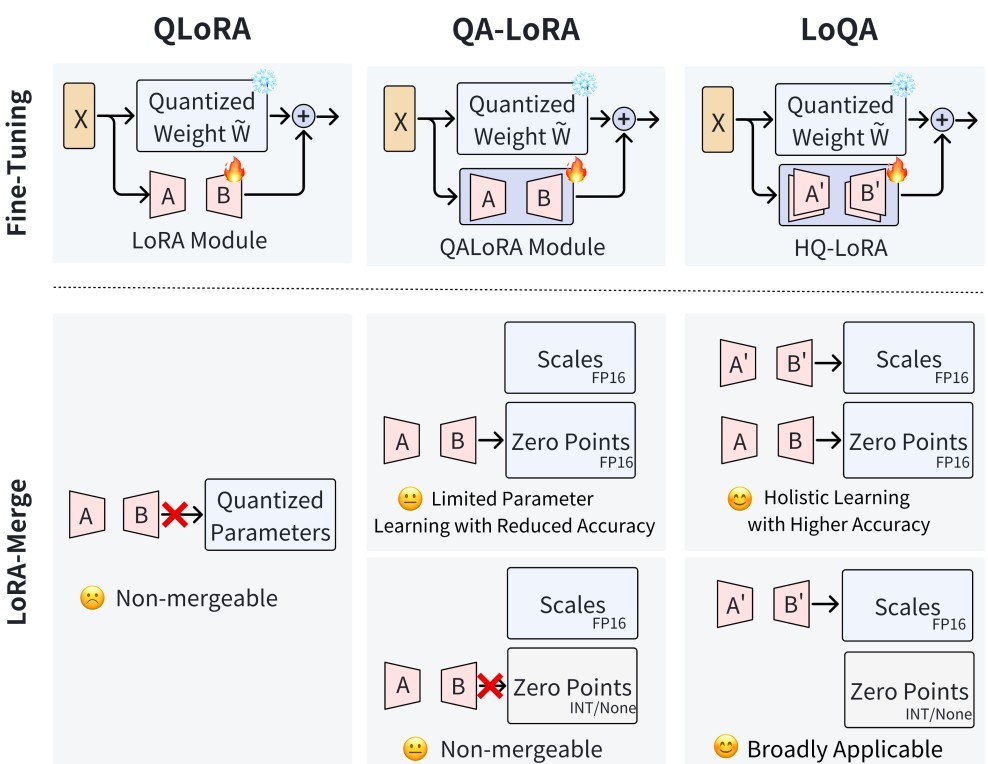

Figure 2: LoQA adopts efficient fine-tuning by applying LoRA after weight quantization. Its core innovation lies in the HQ-LoRA module, which seamlessly integrates LoRA weights into the original quantized weights while maintaining the quantization format, thus preserving inference efficiency. In the "LoRA-Merge" phase, light blue blocks represent components in floating-point format, while light gray blocks indicate non-learnable parts in integer format (see Appendix C). HQ-LoRA resolves QA-LoRA's limitation of non-learnable scales by jointly learning both scales and zero points, achieving superior performance with only half the parameters compared to QA-LoRA (see Table 7). Furthermore, HQ-LoRA enhances generalization capability by enabling scale learning even without zero points or with quantized zero points, reducing dependency on specific PTQ methods.

downstream tasks, we address the limitations of QA-LoRA, which solely adjusts LoRA to learn quantization parameters for zero points $\boldsymbol{Z}$. Instead, we propose an innovative LoRA module that integrates quantized weights $\boldsymbol{W}^{\mathrm{Int}}$, enabling LoRA to learn step sizes $\boldsymbol{S}$. This approach leads to improvements in both generalization and performance. Furthermore, we introduce enhancements to LoRA scaling for quantized parameters, which have demonstrated improved performance across a range of experimental settings.

## 3.1 PRELIMINARY

**Low-Rank Adaptation.** We adopt the symbolic notation system to elucidate the Low-Rank Adaptation (LoRA) methodology (Hu et al., 2021). Let $\boldsymbol{W} \in \mathbb{R}^{D_{\mathrm{out}} \times D_{\mathrm{in}}}$ represent the pretrained weights for a specific layer. Given an input feature vector $\boldsymbol{x} \in \mathbb{R}^{D_{\mathrm{in}}}$, the output vector $\mathbf{y} \in \mathbb{R}^{D_{\mathrm{out}}}$ is computed as $\mathbf{y} = \boldsymbol{W}\boldsymbol{x}$. The LoRA approach introduces two low-rank matrices, $\boldsymbol{A} \in \mathbb{R}^{D_{\mathrm{int}} \times D_{\mathrm{in}}}$ and $\boldsymbol{B} \in \mathbb{R}^{D_{\mathrm{out}} \times D_{\mathrm{int}}}$, where $D_{\mathrm{int}} \ll \min(D_{\mathrm{in}}, D_{\mathrm{out}})$. This ensures that the product $\boldsymbol{B}\boldsymbol{A}$ is a low-rank matrix yet aligns dimensionally with $\boldsymbol{W}$. During training, the computation is augmented with a scaling coefficient $s$:

$$\mathbf{y} = \boldsymbol{W}\boldsymbol{x} + s \cdot \boldsymbol{B}\boldsymbol{A}\boldsymbol{x} \tag{1}$$

We define $\boldsymbol{W}'$ as the final learned weights after fine-tuning, obtained by combining LoRA with the original weights in floating-point format:

$$W' = W + s \cdot BA \tag{2}$$

This formulation allows $W$ to remain static while $A$ and $B$ are updated, enabling efficient parameter tuning. Post-training, we employ the reparametrized weight matrix $W'$ for inference, computing the output as $\mathbf{y} = W'\mathbf{x}$, thus facilitating accelerated computation.

**Joint Low-Rank Adaptation and Quantization.** The integration of quantization and LoRA can further reduce the resource overhead during fine-tuning. First, we quantize the original model. For simplicity, we will use the uniform quantization formula to illustrate this process. To ensure clarity in the following expressions, we will denote the data type in the upper right corner of the different values, using FP16 to represent floating-point numbers. In this paper, we primarily discuss group quantization, so the step sizes $S$ and zero points $Z$ are represented as matrices. If group quantization seems confusing, we provide a detailed explanation of this process using a simple min-max group quantization and dequantization procedure in Appendix C.

$$W^{\text{Int}} = \text{clamp}\left(\lfloor\frac{W^{\text{FP16}} - f(Z^{\text{FP16}}, g)}{f(S^{\text{FP16}}, g)}\rceil, 0, 2^N - 1\right), \tag{3}$$

where $\lfloor\cdot\rceil$ represents the rounding operation. $g$ is the group size for group quantization. $N$ is the final bit number for quantization. The function $f(\mathbf{V}, r)$ is the column duplication operator, which repeats the matrix $\mathbf{V}$ column-wise $r$ times. The detailed definition of $f(\mathbf{V}, r)$ is provided in Appendix E. $S^{\text{FP16}}$ denotes the quantization step size, and $Z^{\text{FP16}}$ serves as the offset or zero point, facilitating the alignment of real and quantized values. After quantization, dequantization is employed during the forward pass to simulate the original weights.

$$\tilde{W}^{\text{FP16}} = W^{\text{Int}} \odot f(S^{\text{FP16}}, g) + f(Z^{\text{FP16}}, g), \tag{4}$$

where $\tilde{W}^{\text{FP16}}$ simulates the original weights, and $\odot$ represents the Hadamard product.

QLoRA (we simply treat the QLoRA quantization process as uniform quantization here) first quantizes the model as in equation 3, then fine-tunes using LoRA (equation 1). The forward process is expressed as:

$$\mathbf{y}' = \tilde{W}^{\text{FP16}}\mathbf{x} + s \cdot BA\mathbf{x} = (W^{\text{Int}} \odot f(S^{\text{FP16}}, g) + f(Z^{\text{FP16}}, g))\mathbf{x} + s \cdot BA\mathbf{x}, \tag{5}$$

where $\mathbf{y}'$ denotes the forward process of the quantized model with LoRA. This fine-tuning process is highly memory-efficient: quantization reduces model weight memory, while LoRA significantly decreases memory required for gradient and optimizer parameters. However, floating-point LoRA cannot be merged into $W^{\text{Int}}$ and can only convert original quantized weights back to floating-point format. QA-LoRA proposes a new LoRA module allowing LoRA weights to be merged into $Z^{\text{FP16}}$, but this limits tunable parameters and yields average performance. Additionally, it struggles to handle situations where $Z^{\text{FP16}}$ is further compressed to $Z^{\text{Int}}$, as discussed in Appendix C.

### 3.2 HOLISTIC QUANTIZATION LOW-RANK ADAPTATION

LoQA retains the prior fine-tuning steps by first quantizing the model and then applying LoRA for fine-tuning, significantly reducing memory consumption for model weights, gradients, and optimizer parameters. During the LoRA fine-tuning phase, we introduce a novel module called HQ-LoRA (Holistic Quantization Low-Rank Adaptation). This module employs two LoRA variants to fine-tune all floating-point quantized parameters within the quantized model (step sizes $S$ and zero points $Z$). This approach allows natural merging of LoRA weights with floating-point parameters from the quantized model post-fine-tuning, without precision loss, ensuring the quantized model maintains its properties. The forward process of HQ-LoRA is expressed as:

$$\mathbf{y}' = \tilde{W}^{\text{FP16}}\mathbf{x} + s \cdot BA\mathbf{x}' + s \cdot (W^{\text{Int}} \odot f((B'A'), g))\mathbf{x}', \tag{6}$$

where $g$ is the group size for group quantization, and $\mathbf{x}'$ is obtained from $\mathbf{x}$ using a grouping operator (one-dimensional average pooling). $A$ and $A'$ are shaped as $D_{\text{int}} \times \frac{D_{\text{in}}}{g}$.

HQ-LoRA's core idea is to align the granularity of all quantization parameters in group quantization with LoRA parameters, ensuring consistent effects for the same input group. We illustrate this with

a simplified example using a quantized model with a group size of 1. The dequantization process becomes $\tilde{\mathbf{W}}^{\text{FP16}} = \mathbf{W}^{\text{Int}} \odot \mathbf{S}^{\text{FP16}} + \mathbf{Z}^{\text{FP16}}$, with $\mathbf{S}$ and $\mathbf{Z}$ shaped $D_{\text{out}} \times D_{\text{in}}$.

The forward formula simplifies to:

$$y' = \tilde{\boldsymbol{W}}^{\text{FP16}}\boldsymbol{x} + s \cdot \boldsymbol{B}\boldsymbol{A}\boldsymbol{x} + s \cdot (\boldsymbol{W}^{\text{Int}} \odot (\boldsymbol{B}'\boldsymbol{A}'))\boldsymbol{x}, \tag{7}$$

In the above equation, $\mathbf{B}'\mathbf{A}'$ can be merged into $\mathbf{S}^{\text{FP16}}$, while $\mathbf{B}\mathbf{A}$ can be merged into $\mathbf{Z}^{\text{FP16}}$. For group sizes $> 1$, we apply one-dimensional average pooling with the corresponding group size to $x$, which reduces the input dimension of $\mathbf{A}$ and $\mathbf{A}'$ to $\frac{D_{\text{in}}}{g}$. This treatment maintains consistency between $\mathbf{B}'\mathbf{A}'$ and $\mathbf{S}^{\text{FP16}}$, and $\mathbf{B}\mathbf{A}$ and $\mathbf{Z}^{\text{FP16}}$ even with group sizes $> 1$, therefore we can still merge $\mathbf{Z}^{\text{FP16}}$ and $\mathbf{B}\mathbf{A}$ after fine-tuning 8.

$$\mathbf{Z}'^{\text{FP16}} = \mathbf{Z}^{\text{FP16}} + s \cdot \mathbf{B}\mathbf{A}, \tag{8}$$

where $\mathbf{Z}'$ represents fine-tuned quantized weights. Similarly, $\mathbf{B}'\mathbf{A}'$ and $\mathbf{S}^{\text{FP16}}$ are merged:

$$\mathbf{S}'^{\text{FP16}} = \mathbf{S}^{\text{FP16}} + s \cdot \mathbf{B}'\mathbf{A}', \tag{9}$$

Here, $\mathbf{S}'$ represents fine-tuned quantized weights. This approach maintains the basic quantization format while incorporating LoRA parameters used during fine-tuning.

### 3.3 QUANTIZED BIT-AWARE SCALING

LoQA introduces Quantized Bit-Aware Scaling (QBAS), a novel approach to adjusting the LoRA scaling factor based on quantization bit-width. In traditional LoRA, the scaling factor $s$ in Equation 1 is defined as $s = \frac{\alpha}{r}$, where $\alpha$ is a hyperparameter and $r$ represents the intermediate dimension size ($D_{\text{int}}$ in Equation 3.1).

Equation 6 reveals that during fine-tuning, while $\boldsymbol{A}$ and $\boldsymbol{B}$ result in a direct update of $s \cdot \boldsymbol{B}\boldsymbol{A}$, the scale-related components $\boldsymbol{A}'$ and $\boldsymbol{B}'$ are modulated by $\boldsymbol{W}_{int}$, yielding an update of $s \cdot \boldsymbol{W}_{int} \odot (\boldsymbol{B}'\boldsymbol{A}')$. Since the magnitude of $\boldsymbol{W}_{int}$ varies with quantization bit-width, it significantly impacts the scale-related LoRA updates.

To address this, QBAS introduces $maxq$ to redesign the LoRA scaling factor as:

$$s = \frac{\alpha}{r \cdot maxq} \tag{10}$$

where $maxq = 2^{N-1}$ and $N$ is the quantization bit-width. As demonstrated in Appendix D, this adjustment effectively normalizes the influence of $\boldsymbol{W}_{int}$, ensuring consistent updates across all layers.

## 4 EXPERIMENTS

### 4.1 MAIN RESULTS

We conducted extensive experiments to assess LoQA's performance in comparison to leading LoRA-finetuning quantization methods, including IR-QLoRA (Qin et al., 2024), QLoRA (Dettmers et al., 2023), and QA-LoRA (Xu et al., 2023). Additionally, we included PEQA (Kim et al., 2023) without LoRA, following the methodology of (Xu et al., 2023). Tables 3 and 1 present the 5-shot accuracy results on the MMLU benchmark (5-shot) after finetuning on the Alpaca (Taori et al., 2023) and Flan v2 (Longpre et al., 2023) datasets, respectively. To ensure fairness, we reproduced the results of QA-LoRA under the same environment and on the same machines for direct comparison. Detailed experimental settings are provided in Appendix B.

**LoQA's performance relative to existing methods:** Our comprehensive analysis reveals that LoQA consistently outperforms comparative quantization methods across various LLaMA model sizes. When compared to the baseline QA-LoRA method, LoQA demonstrates significant accuracy improvements on the MMLU benchmark under identical finetuning conditions. As evidenced in Table 1, the LLaMA-7B model finetuned with LoQA on the Flan v2 dataset achieves an accuracy

Table 1: Accuracy (%) comparison on MMLU benchmark with different quantization methods. Models are finetuned on Flan v2 dataset with rank=64 for all adaptation methods.. **#Bit** denotes bits for weight quantization, where "4+16" indicates LoRA parameters in FP16 that are not mergeable into quantized weights.

| Method | #Bit | MMLU | | | | |
|---|---|---|---|---|---|---|
| | | Hums. | STEM | Social | Other | Avg. |
| LLaMA-7B | 16 | 33.3 | 29.8 | 37.8 | 38.0 | 34.6 |
| NormalFloat | 4 | 33.1 | 30.6 | 38.8 | 38.8 | 35.1 |
| QLoRA w/ GPTQ | 4 | 33.8 | 31.3 | 37.4 | 42.2 | 36.0 |
| QA-LoRA | 4 | 41.8 | 35.6 | 53.7 | 50.8 | 45.2 |
| QLoRA | 4+16 | 41.4 | 35.0 | 49.8 | 52.0 | 44.3 |
| IR-QLoRA | 4+16 | 44.2 | 39.3 | 54.5 | 52.9 | 47.4 |
| **LoQA** | **4** | 43.4 | 37.5 | 56.5 | 53.7 | **47.4** |
| LLaMA-13B | 16 | 40.6 | 36.7 | 48.9 | 48.0 | 43.3 |
| NormalFloat | 4 | 43.0 | 34.5 | 51.8 | 51.4 | 45.0 |
| QLoRA w/ GPTQ | 4 | 48.4 | 38.3 | 54.9 | 55.2 | 49.2 |
| QA-LoRA | 4 | 49.9 | 39.6 | 60.2 | 56.6 | 51.5 |
| QLoRA | 4+16 | 49.9 | 40.1 | 60.2 | 57.9 | 51.9 |
| IR-QLoRA | 4+16 | 49.2 | 41.2 | 62.1 | 59.2 | 52.6 |
| **LoQA** | **4** | 49.2 | 43.3 | 61.6 | 58.8 | **52.9** |
| LLaMA-30B | 16 | 56.2 | 45.9 | 67.1 | 63.9 | 58.2 |
| NormalFloat | 4 | 55.3 | 44.7 | 66.2 | 63.3 | 57.3 |
| QLoRA w/ GPTQ | 4 | 55.8 | 46.4 | 67.0 | 64.0 | 58.1 |
| QA-LoRA | 4 | 55.9 | 47.4 | 69.6 | 65.1 | 59.2 |
| QLoRA | 4+16 | 57.2 | 48.6 | 69.8 | 65.2 | 60.0 |
| IR-QLoRA | 4+16 | 58.1 | 49.4 | 70.7 | 65.8 | 60.8 |
| **LoQA** | **4** | 58.3 | 49.3 | 71.4 | 65.7 | **60.9** |

Table 2: Accuracy (%) comparison of LLaMA under 2-3 bits finetune on the Flan v2 dataset.

| Method | #Bit | MMLU | | | | |
|---|---|---|---|---|---|---|
| | | Hums. | STEM | Social | Other | Avg. |
| LLaMA-7B | 16 | 33.3 | 29.8 | 37.8 | 38.0 | 34.6 |
| NormalFloat | 3 | 30.5 | 29.9 | 34.8 | 34.9 | 32.3 |
| QLoRA w/ GPTQ | 3 | 32.2 | 31.7 | 42.7 | 42.8 | 36.9 |
| QA-LoRA | 3 | 40.8 | 34.7 | 50.5 | 49.8 | 43.7 |
| QLoRA | 3+16 | 41.3 | 37.1 | 50.9 | 49.8 | 44.5 |
| IR-QLoRA | 3+16 | 43.0 | 37.7 | 52.3 | 51.7 | 45.9 |
| **LoQA** | **3** | 43.0 | 38.0 | 55.4 | 51.7 | **46.7** |
| NormalFloat | 2 | 24.2 | 28.9 | 31.1 | 25.0 | 26.9 |
| QLoRA w/ GPTQ | 2 | 23.9 | 25.3 | 26.2 | 25.3 | 25.0 |
| QA-LoRA | 2 | 30.5 | 29.6 | 38.0 | 38.2 | 33.7 |
| QLoRA | 2+16 | 31.8 | 28.7 | 36.7 | 37.7 | 33.5 |
| IR-QLoRA | 2+16 | 31.7 | 29.4 | 37.8 | 36.5 | 33.7 |
| **LoQA** | **2** | 36.7 | 32.7 | 43.3 | 41.4 | **38.4** |

of 47.4%, substantially surpassing the 45.2% accuracy obtained with QA-LoRA. This trend persists in larger models, with LoQA exceeding the baseline by 1.4% and 1.7% for LLaMA-13B and LLaMA-30B, respectively. Moreover, our experiments indicate that LoQA outperforms QLoRA, which employs a combination of 4-bit and 16-bit precision. Notably, LoQA often achieves results comparable to IR-QLoRA, the current SOTA 4+16-bit method, despite operating entirely in a quantized format. This performance parity with higher-precision methods underscores LoQA's efficacy in balancing model compression and task performance.

Table 3: Accuracy (%) comparison of different quantization methods on LLaMA models fine-tuned with Alpaca dataset and evaluated on MMLU.

| Method | #Bit | MMLU | | | | |
|--------|------|------|------|--------|-------|------|
| | | Hums. | STEM | Social | Other | Avg. |
| LLaMA-7B | 16 | 33.3 | 29.8 | 37.8 | 38.0 | 34.6 |
| PEQA | 4 | 34.9 | 28.9 | 37.5 | 40.1 | 34.8 |
| NormalFloat | 4 | 33.1 | 30.6 | 38.8 | 38.8 | 35.1 |
| QLoRA w/ GPTQ | 4 | 33.8 | 31.3 | 37.4 | 42.2 | 36.0 |
| QA-LoRA | 4 | 38.2 | 32.4 | 43.6 | 45.2 | 39.7 |
| QLoRA | 4+16 | 36.1 | 31.9 | 42.0 | 44.5 | 38.4 |
| IR-QLoRA | 4+16 | 38.6 | 34.6 | 45.2 | 45.5 | 40.8 |
| **LoQA** | **4** | 39.0 | 34.2 | 46.2 | 47.5 | **41.5** |

Table 4: Accuracy (%) comparison of LLaMA2 on MMLU. **#Bit** denotes bits for weight quantization, where "4+16" indicates LoRA parameters in FP16 that are not mergeable into quantized weights. The **bold** and underlined numbers represent the best and second-best results respectively.

| Method | Dataset | #Bit | MMLU | | | | |
|--------|---------|------|------|------|--------|-------|------|
| | | | Hums. | STEM | Social | Other | Avg. |
| LLaMA2-7B | - | 16 | 43.0 | 36.4 | 51.4 | 52.2 | 45.5 |
| NormalFloat | - | 4 | 42.0 | 35.9 | 51.0 | 51.4 | 44.8 |
| QA-LoRA | Alpaca | 4 | 42.1 | 34.4 | 49.1 | 50.3 | 43.9 |
| IR-QLoRA | Alpaca | 4+16 | 43.4 | 36.8 | 51.9 | 53.6 | **46.2** |
| **LoQA** | Alpaca | **4** | 41.8 | 38.6 | 51.9 | 53.7 | 46.1 |
| QA-LoRA | Flan v2 | 4 | 45.1 | 39.9 | 58.3 | 56.4 | 49.5 |
| IR-QLoRA | Flan v2 | 4+16 | 49.2 | 41.6 | 60.2 | 58.0 | **52.0** |
| **LoQA** | Flan v2 | **4** | 46.6 | 41.5 | 60.7 | 57.9 | 51.2 |
| LLaMA2-13B | - | 16 | 53.3 | 44.1 | 63.3 | 61.0 | 55.3 |
| NormalFloat | - | 4 | 52.2 | 44.1 | 62.3 | 60.8 | 54.7 |
| QA-LoRA | Alpaca | 4 | 48.0 | 43.0 | 59.7 | 57.4 | 51.7 |
| IR-QLoRA | Alpaca | 4+16 | 51.9 | 43.9 | 61.9 | 60.4 | **54.4** |
| **LoQA** | Alpaca | **4** | 50.9 | 43.8 | 62.9 | 60.6 | 54.2 |
| QA-LoRA | Flan v2 | 4 | 51.2 | 46.2 | 66.9 | 64.3 | **56.6** |
| IR-QLoRA | Flan v2 | 4+16 | 53.1 | 45.6 | 64.9 | 63.8 | 56.5 |
| **LoQA** | Flan v2 | **4** | 52.2 | 46.1 | 66.5 | 62.8 | 56.5 |

**Performance across diverse benchmarks:** To further validate LoQA's effectiveness, we evaluated the models' zero-shot commonsense reasoning capabilities across various tasks. Detailed results of the evaluation, conducted after training on Flan v2 using LLaMA-7B, are presented in Table 5, providing comprehensive evidence of LoQA's consistent and superior performance.

Table 5: Accuracy (%) comparison of 4-bit quantized models on Commonsense QA datasets. Models are evaluated on multiple commonsense reasoning tasks.

| Method | Dataset | CommonsenseQA | | | | | | | |
|--------|---------|-----------|------|------------|-------|-------|-------|------|------|
| | | HellaSwag | PIQA | WinoGrande | ARC-e | ARC-c | BoolQ | OBQA | Avg. |
| LLaMA-7B | - | 56.3 | 78.2 | 67.1 | 67.3 | 38.2 | 72.9 | 28.4 | 58.3 |
| QA-LoRA | Alpaca | 72.2 | 78.9 | 66.3 | 60.9 | 45.1 | 76.9 | 41.0 | 62.9 |
| LoQA | Alpaca | 73.1 | 78.3 | 65.1 | 62.6 | 45.9 | 78.9 | 42.4 | **63.8** |
| QA-LoRA | Flan v2 | 73.6 | 77.6 | 71.4 | 62.1 | 43.2 | 81.7 | 45.2 | 65.0 |
| LoQA | Flan v2 | 73.8 | 78.7 | 71.1 | 63.6 | 44.2 | 82.1 | 45.6 | **65.6** |

**Cross-dataset consistency:** Table 3 presents results obtained using Alpaca (Taori et al., 2023) as the finetuning dataset. Consistent with the Flan v2 dataset results, LoQA consistently achieves optimal

performance, outperforming SOTA methods. This consistency across different finetuning datasets demonstrates LoQA's robustness and generalizability.

**Cross-model generalization:** We extended our analysis to LLaMA2 and LLaMA3 models to assess LoQA's generalization performance across LLM families. Specifically, we applied LoQA to the 7B and 13B variants of LLaMA2 and evaluated their performance on the MMLU benchmark, where LoQA exhibited excellent results. For LLaMA3, LoQA achieved lower training and evaluation loss compared to QA-LoRA, as illustrated in Figure 3, indicating superior data fitting capabilities. However, this did not translate to improved performance on MMLU. As reported in the empirical study by (Huang et al., 2024), LLaMA 3, when converted to NF4 without utilizing any data, achieves a 5-shot accuracy of 62.5 on the MMLU benchmark. However, when fine-tuned on the Alpaca dataset using QLoRA based on the NF4 model, the accuracy decreases to 56.7. We posit that the existing datasets and train-

Table 6: Impact of HQ-LoRA and QBAS on MMLU performance.

| HQ-LoRA | QBAS | #Bit | MMLU |
|:---:|:---:|:---:|:---:|
| ✗ | ✗ | 4 | 45.2 |
| ✗ | ✓ | 4 | 46.2 |
| ✓ | ✗ | 4 | 46.8 |
| ✓ | ✓ | 4 | **47.1** |
| ✗ | ✗ | 3 | 43.7 |
| ✗ | ✓ | 3 | 45.3 |
| ✓ | ✗ | 3 | 44.6 |
| ✓ | ✓ | 3 | **46.7** |
| ✗ | ✗ | 2 | 33.2 |
| ✗ | ✓ | 2 | 35.2 |
| ✓ | ✗ | 2 | 34.8 |
| ✓ | ✓ | 2 | **38.4** |

ing configurations are insufficient to confer positive MMLU benefits to advanced models such as LLaMA 3. Consequently, despite our method's enhanced data fitting capabilities, this advantage does not translate into improved MMLU performance. We contend that this phenomenon warrants further investigation into more suitable datasets and optimized training paradigms.

**LoQA under Ultra-low Bit-width:** We evaluated LoQA's performance under ultra-low bit-width conditions and compared it with other SOTA methods. Table 2 demonstrates LoQA's superior performance in this domain. Notably, the 2-bit LoQA configuration outperforms the current 2+16-bit SOTA method IR-QLoRA by 4.7%. Furthermore, even in its 2-bit configuration, LoQA surpasses the original 16-bit model by 3.8%, highlighting its exceptional efficiency in low-bit scenarios.

## 4.2 ABLATION ANALYSIS

To elucidate the efficacy of the techniques employed in LoQA on both accuracy and efficiency, we conducted comprehensive ablation studies using the LLaMA-7B model on the Flan v2 dataset.

**Accuracy Ablation:** We performed ablation experiments on our proposed HQ-LoRA and QBAS methods to assess their individual contributions. Given that QBAS involves different bit-widths, we examined 4-bit, 3-bit, and 2-bit configurations, testing various combinations of the two methods. As illustrated in Table 6, both HQ-LoRA and QBAS prove crucial for performance optimization. The synergistic combination of these methods yields the most superior performance.

**Trainable Parameters Ablation:** LoQA utilizes HQ-LoRA to adjust all tunable parameters in the quantized model, effectively doubling the number of learnable parameters compared to QA-LoRA. We conducted ablation experiments by varying the rank to investigate the impact of different quantities of learnable parameters. We conducted a series of experiments exploring different ranks of LoRA on the Flan v2 dataset. Our findings indicate that increasing the rank by multiples generally did not yield substantial performance improvements. Notably, HQ-LoRA achieved superior results even with half the number of parameters, as illustrated in Table 7. This observation suggests that the efficacy of

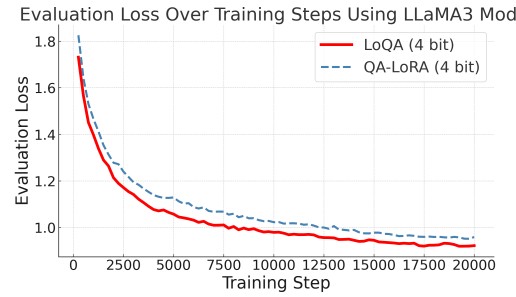

Figure 3: Evaluation loss trajectories for LoQA and QA-LoRA applied to the LLaMA3 model

our proposed HQ-LoRA method is not solely dependent on the number of trainable parameters, but rather on its intrinsic ability to more efficiently utilize the parameter space.

## 4.3 DISCUSSION

**Larger Quantization Group Size:** To provide a more comprehensive evaluation of our method's effectiveness, we conducted experiments using a group size of 128 in the FLAN-v2 dataset. Table 8 presents the results, which demonstrate the robustness and efficacy of our proposed approach under this larger group size configuration. These findings suggest that our method maintains its performance advantages even when scaling to larger quantization groups, indicating its potential applicability across various quantization settings.

**Training Cost:** As shown in Appendix G, LoQA requires approximately 1.3 times the training time of QA-LoRA, while LoQA-S demands even less than this 1.3-fold increase. For context, according to the QA-LoRA study, QLoRA necessitates approximately twice the training time of QA-LoRA. These results indicate that, under equivalent optimization conditions, LoQA achieves optimal results with a balanced training cost.

**Inference Efficiency:** HQ-LoRA's flexibility in selecting learnable quantized weights and reparameterizing the learned parameters into the original quantized model weights enables us to achieve inference speeds comparable to other weight-only quantization models. Our approach is compatible with various acceleration toolboxes, including MLC-LLM (team, 2023), AWQ (Lin et al., 2023), BitBLAS (Team), and Marlin (Frantar et al., 2024). As shown in Appendix H, we provide inference speed benchmarks using Marlin on A100 GPU.

Table 7: Accuracy (%) comparison of LLaMA with different parameter scales on MMLU 5-shot tasks, evaluating the performance between LoQA and QA-LoRA under various rank settings

| Method | Rank | #Bit | MMLU | | | | |
| --- | --- | --- | --- | --- | --- | --- | --- |
| | | | Hums. | STEM | Social | Other | Avg. |
| LLaMA-7B | - | 16 | 33.3 | 29.8 | 37.8 | 38.0 | 34.6 |
| QA-LoRA | 64 | 4 | 41.8 | 35.6 | 53.7 | 50.8 | 45.2 |
| QA-LoRA | 128 | 4 | 42.6 | 35.8 | 53.3 | 51.5 | 45.5 |
| HQ-LoRA (ours) | 32 | 4 | 44.0 | 36.9 | 56.5 | 51.3 | 46.9 |
| HQ-LoRA (ours) | 64 | 4 | 44.0 | 37.2 | 56.1 | 52.3 | **47.1** |
| QA-LoRA | 64 | 2 | 30.5 | 29.6 | 38.0 | 38.2 | 33.7 |
| QA-LoRA | 128 | 2 | 29.7 | 29.3 | 36.8 | 35.9 | 32.6 |
| HQ-LoRA (ours) | 32 | 2 | 32.2 | 31.3 | 41.3 | 40.2 | 35.8 |
| HQ-LoRA (ours) | 64 | 2 | 34.2 | 28.8 | 41.0 | 40.5 | **36.0** |

Table 8: Accuracy (%) comparison on MMLU 5-shot benchmark with group size 128.

| Method | Dataset | Humanities | STEM | Social Sciences | Other | Avg. |
| --- | --- | --- | --- | --- | --- | --- |
| Llama-2-7B | - | 43.0 | 36.4 | 51.4 | 52.2 | 45.5 |
| QA-LoRA | Alpaca | 42.6 | 36.8 | 50.0 | 50.6 | 44.8 |
| HQ-LoRA (ours) | Alpaca | 43.3 | 36.6 | 51.4 | 52.6 | **45.8** |
| QA-LoRA | FLAN v2 | 44.9 | 39.7 | 57.9 | 56.2 | 49.2 |
| HQ-LoRA (ours) | FLAN v2 | 48.2 | 40.8 | 60.7 | 58.5 | **51.7** |

## 5 CONCLUSION

In this study, we introduce LoQA, a novel approach that incorporates HQ-LoRA for effective fine-tuning of all quantized parameters. We also developed QBAS, an innovative LoRA scaling strategy capable of adjusting the scaling size based on the quantization bit-width. This approach demonstrates flexibility in its application to various uniform quantization methods, offering a robust solution for efficient model adaptation and deployment.

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

## A  LIMITATIONS

As elucidated in Section 4.1, the base LoRA, datasets, and training methodologies employed in this study are not reflective of the most current advancements in the field. These components necessitate further refinement to achieve optimal results. However, due to resource constraints, we are unable to replicate all previous work using the latest datasets and training paradigms. Consequently, we present our approach under conditions where extraneous variables are controlled to the greatest extent possible, ensuring a fair comparison within the constraints of our experimental setup.

## B  SETTINGS

**Foundation models and quantization detial.** We conducted a series of experiments utilizing the LoQA framework on various models from the LLaMA series, including LLaMA (Touvron et al., 2023a), LLaMA2 (Touvron et al., 2023b), and LLaMA3. Our experimental setup encompassed base models such as the 7B, 13B, and 33B configurations from LLaMA, the 7B and 13B models from LLaMA2, and the 8B model from LLaMA3. In the quantization step, we employed a Post-Training Quantization method named GPTQ (Frantar et al., 2023) and LoQA extensively supports other Post-Training Quantization (PTQ). We used the same GPTQ settings for model quantization between different methods. In our main experiments, we implemented group-wise asymmetric quantization (with a group size of 32). We set the 'desc_act' variable to false and the 'true-sequential' variable to true, and the calibration dataset is wikitext2.

**Evaluation metrics.** In alignment with recent methodologies (Xu et al., 2023),(Dettmers et al., 2023), we evaluated the zero-shot and few-shot performance of these large language models (LLMs) on the Massively Multitask Language Understanding (MMLU) benchmark (Hendrycks et al., 2021). This benchmark encompasses 57 language tasks across fields like humanities, STEM, and social sciences. We utilized the official MMLU evaluation script and prompts. Furthermore, we assessed the models' zero-shot common sense reasoning abilities on tasks such as HellaSwag (Zellers et al., 2019), PIQA (Bisk et al., 2020), WinoGrande (Sakaguchi et al., 2019), ARC (Clark et al., 2018), BoolQ (Clark et al., 2019), and OpenBookQA (Mihaylov et al., 2018). The 'lm-eval' tool (Gao et al., 2021) was used to generate the Common Sense QA results, and we consistently used the final checkpoint's results for evaluation.

**Datasets and Training Details.** For our fine-tuning datasets, we selected Alpaca (Taori et al., 2023) and FLAN v2 (Longpre et al., 2023). Alpaca contains 52K instruction-following data generated from text-davinci-003 (GPT 3.5) (Wang et al., 2022), and was trained for 10k steps. FLAN v2 is a collection of 1,836 tasks combining CoT, Muffin, T0-SF, and NIV2. In accordance with previous work, we used a batch size of 16, and FLAN v2 was trained for 20k steps on a randomly selected 320K subset used for training. To ensure a fair comparison, we maintained consistency in training hyperparameters with previous studies. All our experiments are conducted on Nvidia Tesla A100 GPUs.

Table 9: Key Training Parameters and Values

| Parameter | Value |
|---|---|
| Learning Rate | 0.0002 |
| Batch Size per GPU | 1 |
| Gradient Accumulation Steps | 16 |
| Weight Decay | 0.0 |
| LoRA Rank | 64 |
| LoRA Alpha | 16 |
| LoRA Dropout | 0.0 |
| Gradient Checkpointing | True |
| Warmup Ratio | 0.03 |
| Learning Rate Scheduler Type | constant |

## C  QUANTIZATION AND DEQUANTIZATION.

### C.1  GROUP-WISE QUANTIZATION

Quantization can be implemented at various levels of granularity, commonly categorized into per-tensor, per-channel, and group quantization. In the most coarse-grained scenario, per-tensor quantization, the entire weight matrix $\mathbf{W}^{\text{FP16}}$ utilizes a single quantization step size ($s$) and zero point ($z$). This section will first introduce per-tensor quantization and dequantization, followed by an explication of the distinctions between group quantization and per-tensor quantization.

To elucidate this concept, we will examine the application of a standard min-max quantization method. Consider a model with weights in FP16 format (denoted as $\boldsymbol{W}^{\text{FP16}}$), which we aim to quantize to $N$ bits. The quantization process is governed by the following formulation. For consistency, we will uniformly denote floating-point numbers with the superscript FP16.

$$
\begin{aligned}
\boldsymbol{W}^{\text{Int}} &= \text{clamp}\left(\lfloor\frac{\boldsymbol{W}^{\text{FP16}} - z^{\text{FP16}}}{s^{\text{FP16}}}\rceil, 0, 2^N - 1\right), \\
z^{\text{FP16}} &= \boldsymbol{W}^{\text{FP16}}_{\min}, \\
s^{\text{FP16}} &= \frac{\boldsymbol{W}^{\text{FP16}}_{\max} - \boldsymbol{W}^{\text{FP16}}_{\min}}{2^N - 1}.
\end{aligned}
\tag{11}
$$

Here, $\lfloor\cdot\rceil$ denotes the rounding operation, $N$ represents the target bit number, $s^{\text{FP16}}$ is the quantization step size, and $z^{\text{FP16}}$ serves as the offset or zero point. The function $\text{clamp}(z, r_1, r_2)$ constrains the value of $z$ within the range defined by $r_1$ and $r_2$, effectively bounding it by returning $r_1$ if $z$ is less than $r_1$, and $r_2$ if $z$ exceeds $r_2$.

This quantization procedure involves storing the values of $\boldsymbol{W}^{\text{Int}}$, $z^{\text{FP16}}$, and $s^{\text{FP16}}$. To revert to the floating-point representation $\boldsymbol{W}^{\text{FP16}}$ during inference, we employ the corresponding dequantization process:

$$
\tilde{\boldsymbol{W}}^{\text{FP16}} = \boldsymbol{W}^{\text{Int}} s^{\text{FP16}} + z^{\text{FP16}},
\tag{12}
$$

where $\tilde{\boldsymbol{W}}^{\text{FP16}}$ serves as an approximation of the original weight matrix $\boldsymbol{W}^{\text{FP16}}$. This approximation facilitates the restoration of floating-point values from their quantized integer form, enabling the use of lightweight models in high-precision tasks.

## C.2 ZERO-POINT COMPRESSION IN PTQ

Recent advancements in Post-Training Quantization (PTQ) have focused on optimizing memory efficiency for Large Language Model (LLM) inference, which is primarily memory-bounded. Weight-only quantization methods accelerate computation by reducing memory access, and many widely-used PTQ methods have introduced innovative approaches to handle zero points. These methods either compress zero points to integers (as seen in OmniQuant (Shao et al., 2023), AffineQuant (Ma et al., 2024) or eliminate them entirely (as demonstrated in SmoothQuant (Xiao et al., 2023) and AWQ (Lin et al., 2023)). This trend is further reinforced by acceleration libraries like Marlin, which specifically do not support floating-point zero points.

The quantization procedure for methods employing integer zero points typically follows:

$$
\begin{aligned}
\boldsymbol{W}^{\text{Int}} &= \text{clamp}\left(\text{round}\left(\frac{\boldsymbol{W}^{\text{FP16}}}{s^{\text{FP16}}} - z^{\text{Int}}\right), 0, 2^N - 1\right), \\
z^{\text{Int}} &= \text{round}\left(\frac{\boldsymbol{W}^{\text{FP16}}_{\min}}{s^{\text{FP16}}}\right), \\
s^{\text{FP16}} &= \frac{\boldsymbol{W}^{\text{FP16}}_{\max} - \boldsymbol{W}^{\text{FP16}}_{\min}}{2^N - 1}.
\end{aligned}
\tag{13}
$$

The corresponding dequantization process is described as follows:

$$
\tilde{\boldsymbol{W}}^{\text{FP16}} = (\boldsymbol{W}^{\text{Int}} - z^{\text{Int}}) s^{\text{FP16}}
\tag{14}
$$

In this quantization framework, both $\mathbf{W}^{\text{FP16}}$ and $\mathbf{W}^{\text{Int}}$ matrices are dimensioned as $D_{\text{out}} \times D_{\text{in}}$. The quantization parameters vary in structure depending on the granularity level: for per-tensor quantization, $s^{\text{FP16}}$ and $z^{\text{Int}}$ are scalars, while for per-channel or group quantization, they become vectors or matrices. In group quantization, parameters within each row are divided into groups of fixed size, with each group sharing a single $s$ and $z$. This results in quantization parameters $\mathbf{S}$ and $\mathbf{Z}$ with dimensions $D_{\text{out}} \times \frac{D_{\text{in}}}{\text{groupsize}}$, following the same format as equation 3.

# D ANALYSIS OF LORA MAGNITUDE WITH QBAS

We conducted a statistical analysis of LoRA magnitudes with and without QBAS on LLaMA-7B (4-bit) fine-tuned on the Flan v2 dataset. Our analysis reveals that QBAS effectively regulates the

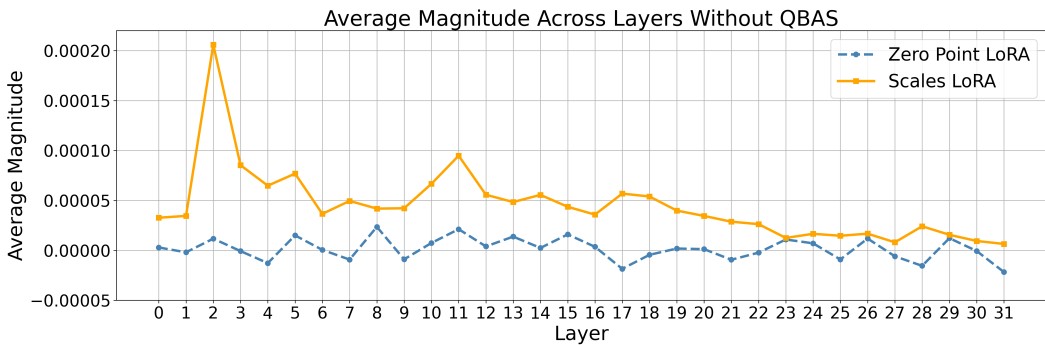

Figure 4: Distribution of LoRA magnitude across layers without QBAS. Experiments conducted on 4-bit quantized LLaMA-7B fine-tuned on Flan v2 dataset.

influence of $\boldsymbol{W}_{int}$ on the scale-related LoRA parameters. Specifically, QBAS helps maintain reasonable magnitudes of LoRA updates across different layers.

As shown in Figure 4 and 5, the analysis reveals that without QBAS, the LoRA magnitudes tend to be excessively large with high inter-layer variance. QBAS significantly reduces both the absolute magnitude and the layer-wise variance of LoRA updates, leading to more controlled parameter adjustments during the learning process.

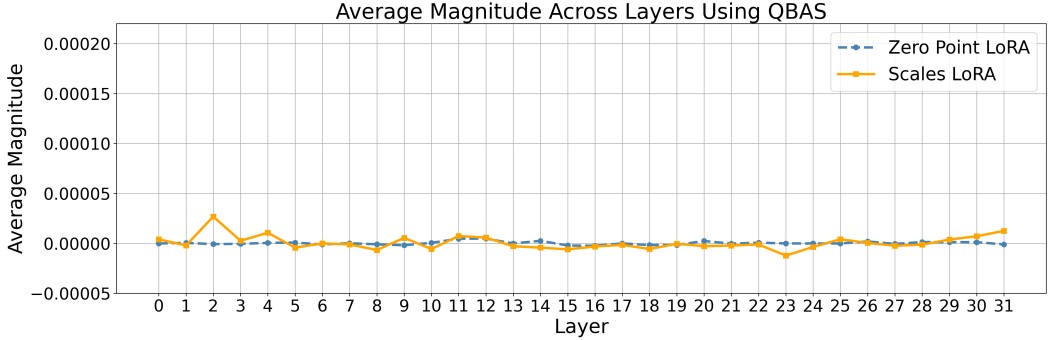

Figure 5: Distribution of LoRA magnitude across layers with QBAS. Experiments conducted on 4-bit quantized LLaMA-7B fine-tuned on Flan v2 dataset.

## E   DEFINITION OF THE COLUMN DUPLICATION OPERATOR

Assume we have a matrix $\mathbf{V} \in \mathbb{R}^{m \times n}$, where $\mathbf{V} = [\mathbf{v}_1, \mathbf{v}_2, \ldots, \mathbf{v}_n]$, and each $\mathbf{v}_i \in \mathbb{R}^m$ represents a column vector. We want to define an operator $f$ that repeats each column vector $\mathbf{v}_i$ exactly $r$ times along the second dimension.

The resulting matrix, denoted by $f(\mathbf{V}, r)$, will then have dimensions $m \times (nr)$ and can be expressed as:

$$f(\mathbf{V}, r) = [\underbrace{\mathbf{v}_1, \ldots, \mathbf{v}_1}_{r \text{ times}}, \underbrace{\mathbf{v}_2, \ldots, \mathbf{v}_2}_{r \text{ times}}, \ldots, \underbrace{\mathbf{v}_n, \ldots, \mathbf{v}_n}_{r \text{ times}}] \tag{15}$$

## F   ADDITIONAL EXPERIMENTS ON OPT-6.7B

To validate the effectiveness of LoQA beyond the LLaMA family, we conduct experiments on OPT-6.7B using the Flan v2 dataset. The results demonstrate that LoQA significantly outperforms QA-

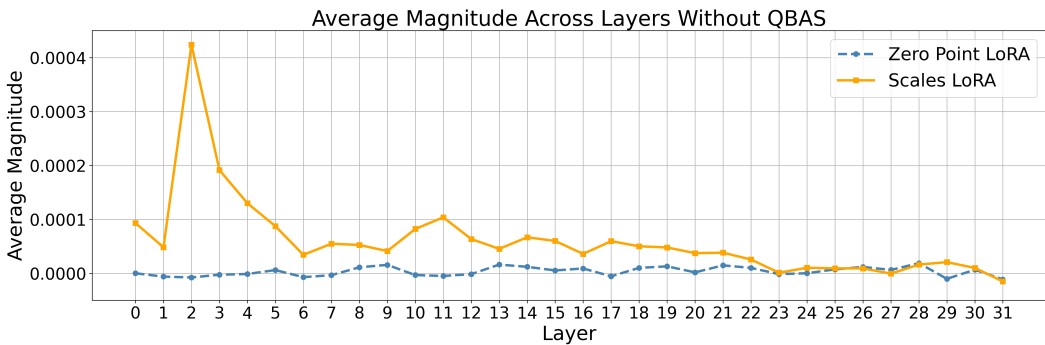

Figure 6: Distribution of LoRA magnitude across layers without QBAS. Experiments conducted on 3-bit quantized LLaMA-7B fine-tuned on Flan v2 dataset.

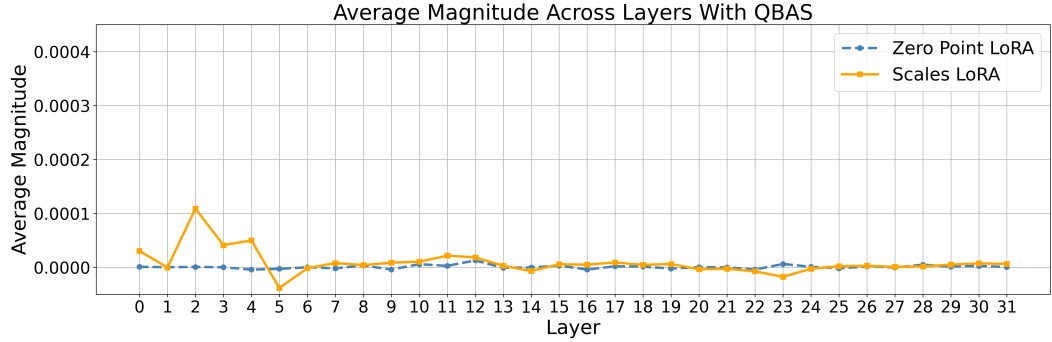

Figure 7: Distribution of LoRA magnitude across layers with QBAS. Experiments conducted on 3-bit quantized LLaMA-7B fine-tuned on Flan v2 dataset.

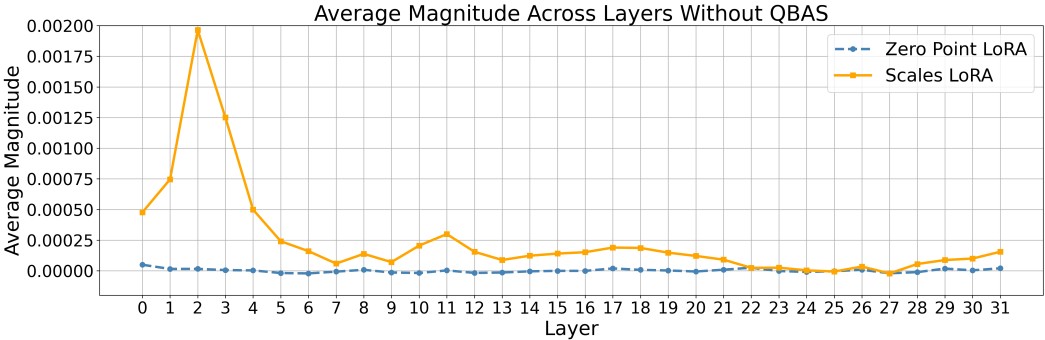

Figure 8: Distribution of LoRA magnitude across layers without QBAS. Experiments conducted on 2-bit quantized LLaMA-7B fine-tuned on Flan v2 dataset.

LoRA on the MMLU benchmark with 5-shot prompting, achieving 33.8% accuracy compared to QA-LoRA's 29.8%.

## G    TRAINING SPEED TEST SETTINGS

We evaluated the time and memory consumption of LoQA and QA-LoRA, both implemented with PyTorch backend, under identical environmental conditions and hardware configurations. We performed measurements of both time and memory usage on the Flan v2 dataset, ensuring consistent machine and environmental conditions for all experiments. In our notation, LoQA-S represents the

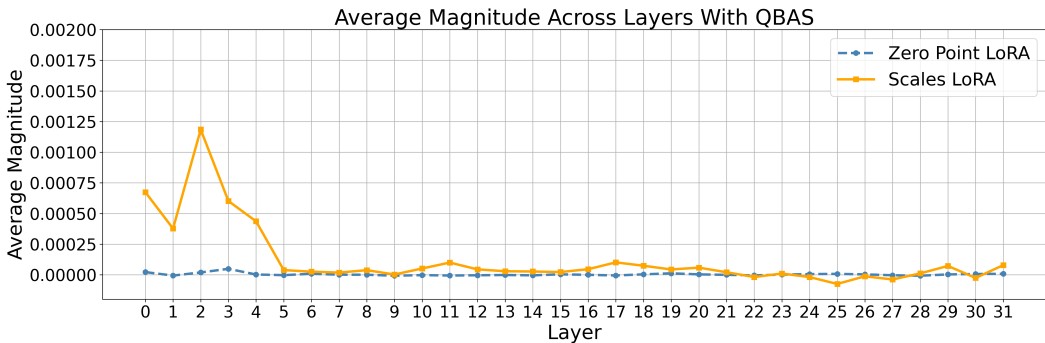

Figure 9: Distribution of LoRA magnitude across layers with QBAS. Experiments conducted on 2-bit quantized LLaMA-7B fine-tuned on Flan v2 dataset.

Table 10: Accuracy (%) comparison on MMLU 5-shot benchmark using OPT-6.7B. All methods are trained on Flan v2 dataset.

| Method | #Bit | Humanities | STEM | Social Sciences | Other | Avg. |
|--------|------|-----------|------|-----------------|-------|------|
| OPT 6.7B | 16 | - | - | - | - | 24.6 |
| QA-LoRA | 4 | 27.2 | 30.3 | 33.3 | 30.1 | 29.8 |
| LoQA | 4 | 29.4 | 31.2 | 40.1 | 36.6 | **33.8** |

scenario where only the quantization parameter is adjusted. And appendix provides the detailed experimental setup for the training speed tests. The experiments were conducted to compare the training efficiency of the LLaMA-7B model at different quantization levels (bit precision) on the Flan v2 dataset. The settings are as follows:

- **Model:** LLaMA-7B
- **Dataset:** Flan v2 dataset with a total of 320k examples.
- **Quantization Bits:** Various bit precisions (e.g., 4-bit, 3-bit, 2-bit) were evaluated to observe their impact on training speed.
- **Hardware:** Eight NVIDIA RTX 3090 GPUs were used for all experiments.
- **Framework:** PyTorch was used as the backend for model training and computation.
- **Environment:** The experiments were conducted in the same environment as QA-LoRA to ensure fair comparisons. The setup included the same data preprocessing pipeline, optimizer, and learning rate scheduler as used in QA-LoRA.

The training speed was measured by recording the average number of training steps per second for each quantization level. This comparison highlights the trade-offs between computational efficiency and precision during model training.

## H    INFERENCE SPEED TEST SETTINGS

This appendix provides detailed information about the settings used for the inference speed tests described in the main text. The results in Table 12 were obtained using the following setup:

- **Framework:** Marlin (Frantar et al., 2024)
- **Hardware:** NVIDIA A100 GPU
- **Batch size:** 16
- **Group size:** 128
- **Quantization:** Zero Point quantization was applied.

Table 11: Comparison of Training Time and Memory Usage across Different Models

| llama-7B-w4a16g32 | LoQA | LoQA-S | QA-LoRA |
|---|---|---|---|
| **Training Time (h)** | 21 | 17 | 16 |
| **Memory (GB)** | 12.0 | 11.6 | 10.8 |
| **llama-7B-w3a16g32** | **LoQA** | **LoQA-S** | **QA-LoRA** |
| **Training Time (h)** | 26 | 23 | 21 |
| **Memory (GB)** | 12.0 | 11.6 | 10.8 |
| **llama-7B-w2a16g32** | **LoQA** | **LoQA-S** | **QA-LoRA** |
| **Training Time (h)** | 21 | 18 | 16 |
| **Memory (GB)** | 10.3 | 10.0 | 9.4 |

The speedup values in the table demonstrate the benefits of Marlin's optimizations and the efficiency of Zero Point quantization for large language model inference on high-performance GPUs.

Table 12: Performance comparison of models in terms of TFLOP/s and speedup. The speedup results were obtained using Marlin on an NVIDIA A100 GPU.

| Model | TFLOP/s | Speedup |
|---|---|---|
| Llama7B | 63.788 | 2.71 |
| Llama13B | 76.907 | 3.31 |
| Llama33B | 87.907 | 3.50 |
| Llama65B | 92.807 | 3.61 |
| Falcon180B | 104.5 | 3.81 |

