# OpenReview forum: "Low Rank Quantization Adaptation for Large Language Model"
_ICLR.cc/2025/Conference — Submitted to ICLR 2025_

### Official Review · Reviewer_2qNk · 2024-10-28

**Soundness:** 3
**Presentation:** 2
**Contribution:** 2
**Rating:** 5
**Confidence:** 4

**Summary:**

This paper introduces Low Rank Quantization Adaptation (LoQA) for LLMs, a novel parameter-efficient fine-tuning method for quantized models. LoQA aims to reduce memory consumption across model weights, gradients, and optimizer parameters. It includes two core modules: **HQ-LoRA** (Holistic Quantization Low-Rank Adaptation) and **QBAS** (Quantized Bit-Aware Scaling), aligning quantization parameter granularity with LoRA parameters in group quantization.

**Strengths:**

- The proposed method is well-motivated and sounded.
- This approach is straightforward yet effective, especially in ultra-low bit-width scenarios.
- Code is provided for reproducibility.

**Weaknesses:**

- **Limited novelty**: The method’s innovation is incremental, building on QA-LoRA and closely resembling group quantization applied to LoRA parameters. In Sec 3.2, LoQA uses two LoRA components, whereas QA-LoRA requires only one, which may lead to a higher parameter count.
- **Experimental limitations**:
  - LoQA does not demonstrate clear improvements on LLaMA 2, and for LLaMA 3, only training loss rather than performance results is presented. Providing promising results on more advanced models would strengthen the paper’s contribution.
  - Important baselines, such as **IR-QLoRA** and **QLoRA**, are absent in Tables 5,7,9, affecting comparison fairness.
  - While the authors claim broad applicability of LoQA across post-training quantization techniques, only **GPTQ** settings are evaluated.
- **Poor presentation and writing**:
  - Figure 2 lacks clarity in distinguishing between QA-LoRA and LoQA.
  - In Sec 3.3, the rationale for setting $maxq$ to $2^{N-1}$ is not sufficiently analyzed.
  - In some tables (e.g., Tables 4, 7, 8), entries are missing bold formatting to indicate the best results.
  - The LoRA rank used in the compared methods is not clearly specified.
  - The "4+16 bit" setting is not formally defined.
  - "Llama-7b" and "Llama-7B" are inconsistently referenced throughout the paper.
  - "Post-fine-tuning" in line 269 is incomplete.

**Questions:**

Please refer to the Weaknesses above for questions.

---

> ### Author Response · Authors · 2024-11-20
>
> Thank you for your thorough and constructive feedback.
>
> > • **Limited novelty**: The method’s innovation is incremental, building on QA-LoRA and closely resembling group quantization applied to LoRA parameters. In Sec 3.2, LoQA uses two LoRA components, whereas QA-LoRA requires only one, which may lead to a higher parameter count.
> >
>
> We respectfully disagree with the characterization of LoQA as merely incremental. Like QA-LoRA, our method LoQA employs LoRA and supports reparameterization of LoRA into quantized weights. However, LoQA makes two fundamental contributions beyond QA-LoRA: (1) it pioneers the novel concept of merging LoRA into scale quantization parameters, and (2) introduces QBAS to address the abnormal magnitude issues in learning scale parameters. These advances represent significant improvements in both versatility and accuracy, supported by the following evidence:
>
> 1. Firstly, LLM inference is primarily memory-bounded, where weight-only quantization methods accelerate computation by reducing memory access. Most widely-used PTQ methods compress zero points to integers (e.g., Omniquant[1], AffineQuant[2], FlatQuant[3]) or eliminate them entirely (e.g., Smooth Quant[4], AWQ[5]) for further acceleration. Additionally, some acceleration libraries like Marlin do not support floating-point zero points. Therefore, our innovation of introducing LoRA to scale parameters has substantial practical implications for deploying these methods.
>
>     [1] Shao W, Chen M, Zhang Z, et al. Omniquant: Omnidirectionally calibrated quantization for large language models[J]. arXiv preprint arXiv:2308.13137, 2023.
>
>     [2]Ma Y, Li H, Zheng X, et al. Affinequant: Affine transformation quantization for large language models[J]. arXiv preprint arXiv:2403.12544, 2024.
>
>     [3]Sun Y, Liu R, Bai H, et al. FlatQuant: Flatness Matters for LLM Quantization[J]. arXiv preprint arXiv:2410.09426, 2024.
>
>     [4]Xiao G, Lin J, Seznec M, et al. Smoothquant: Accurate and efficient post-training quantization for large language models[C]//International Conference on Machine Learning. PMLR, 2023: 38087-38099.
>
>     [5]Lin J, Tang J, Tang H, et al. AWQ: Activation-aware Weight Quantization for On-Device LLM Compression and Acceleration[J]. Proceedings of Machine Learning and Systems, 2024, 6: 87-100.
>
> 2. Next, as demonstrated in our ablation studies (Section 4.2), when both zero points and scales are learnable, learning scale parameters achieves significantly better accuracy than increasing LoRA rank, despite using only half the number of parameters. This substantial improvement in accuracy with fewer parameters clearly demonstrates the effectiveness of our approach to scale parameter learning. Extensive experiments demonstrate that LoQA consistently outperforms previous fine-tuning methods that maintain quantized formats, and in many cases, matches the performance of state-of-the-art 4+16 bit methods. Notably, in ultra-low bit-width scenarios, LoQA's effectiveness is even more pronounced, with its 2-bit version surpassing the current 2+16-bit state-of-the-art method by 4.7\% and even outperforming the original 16-bit model.

---

> ### Author Response · Authors · 2024-11-20
>
> > LoQA does not demonstrate clear improvements on LLaMA 2, and for LLaMA 3, only training loss rather than performance results is presented. Providing promising results on more advanced models would strengthen the paper’s contribution.
> >
>
> There appears to be a misunderstanding regarding our experimental results:
>
> First, regarding LLaMA 2 results (Tables 4 and 9), it's crucial to note that LoQA achieves its performance with true 4-bit models, while IR-QLoRA and QLoRA operate at 4+16 bits since their LoRA weights cannot be merged into quantized weights. Our method achieves comparable performance to these 4+16-bit models while maintaining quantized format. Compared to previous 4-bit methods, LoQA shows clear improvements across most tasks.
>
> Second, for LLaMA 3, we specifically present validation loss, not training loss, demonstrating our method's strong generalization capability on downstream tasks. This validation performance indicates that LoQA effectively adapts to newer model architectures while maintaining good fitting characteristics, as we've also detailed in our response to Reviewer n8i8.
>
> > Important baselines, such as **IR-QLoRA** and **QLoRA**, are absent in Tables 5,7,9, affecting comparison fairness.
> >
>
> For Table 5, we have added comprehensive comparison results on the CommonsenQA dataset using Alpaca data:
>
> | Method | #Bit | Avg. |
> | --- | --- | --- |
> | LLaMA-7B | 16 | 58.3 |
> | NormalFloat | 4 | 61.6 |
> | QLoRA w/ GPTQ | 4 | 59.8 |
> | QA-LoRA | 4 | 61.8 |
> | QLoRA | 4+16 | 62 |
> | IR-QLoRA | 4+16 | 63.7 |
> | LoQA | 4 | 63.8 |
>
> Regarding Table 7, while it doesn't directly show comparisons with other methods, we provide detailed comparative analysis and discussion in Section 4.3.
>
> Table 9 specifically focuses on ablation studies across different group sizes and datasets with QA-LoRA, as its primary purpose is to demonstrate our method's effectiveness under varying group size configurations, rather than comprehensive baseline comparisons.
>
> > While the authors claim broad applicability of LoQA across post-training quantization techniques, only **GPTQ** settings are evaluated.
> >
>
> The consistent use of GPTQ across all methods in Tables 1, 2, and 3 was chosen to ensure fair comparison. QLoRA uses GPTQ to recover 4-bit format, and QA-LoRA uses GPTQ as its fine-tuning starting point. Therefore, LoQA maintains this consistency for meaningful comparative evaluation.
>
> From a theoretical perspective, LoQA's applicability extends well beyond GPTQ. Our method naturally supports various PTQ algorithms, particularly those with integer zero points or zero point-free approaches, as LoRA can be reparameterized into the scale component. This broader compatibility can be formally demonstrated through theoretical analysis, though we focused our experimental evaluation on GPTQ to maintain consistent comparison conditions.
>
> > In Sec 3.3, the rationale for setting maxq is not sufficiently analyzed.
> >
>
> We have enhanced the explanation of maxq in Section 3.3 and provided comprehensive visual analysis in Appendix D. The appendix now includes visualization that clearly demonstrates the effect of QBAS and helps readers intuitively understand the rationale behind our maxq setting. These visualizations effectively illustrate how QBAS addresses scale parameter learning challenges.
>
> > Figure 2 lacks clarity in distinguishing between QA-LoRA and LoQA.
> >
> >
> > "Post-fine-tuning" in line 269 is incomplete.
> >
> > "Llama-7b" and "Llama-7B" are inconsistently referenced throughout the paper.
> >
>
> Thank you for these detailed observations. We have addressed each point:
>
> 1. We have enhanced Figure 2 to more clearly differentiate between QA-LoRA and LoQA, with improved labeling and visual distinction between their respective architectures.
> 2. We have expanded the "post-fine-tuning" discussion in line 269 to provide complete context and implementation details.
> 3. We have standardized all model references to "LLaMA-7B" throughout the paper for consistency.
>
> > The "4+16 bit" setting is not formally defined.
> >
>
> As described in Section 3.1, the "4+16 bit" notation represents models where the backbone network is quantized to 4 bits while LoRA parameters remain in 16-bit precision, as used in QLoRA and IR-QLoRA. In contrast, QA-LoRA and LoQA achieve true 4-bit models by merging the LoRA parameters into the quantized weights. We have added explicit clarification of this notation in all relevant table captions to prevent any confusion.

---

> > ### Comment · Reviewer_2qNk · 2024-11-22
> >
> > Dear Authors,
> >
> > Thank you for your detailed response. Most of the unclear parts in the paper have been clarified, but I still have a few concerns:
> >
> > - The LoRA-Merge section in **Figure 2 is still not clear**. I suggest that the authors enhance the caption to describe the meaning of each block in the figure.
> > - The paper has undergone significant modifications. I recommend that the authors **highlight the changes in a different color** to better showcase these updates or provide a detailed explanation of the revisions in the general response.
> > - I am still concerned about the novelty: Regarding the significance of maxq, Appendix D provides experiments for the 4-bit case. **Are there any analyses for different bit-widths?** Additionally, could the distribution of LoRA magnitude also be influenced by initialization or learning rate? Please provide some analysis to further clarify the importance of QBAS.
> >
> > If these issues can be addressed, I would be happy to consider raising my score.

---

> ### Author Response · Authors · 2024-11-23
>
> Dear Reviewer,
>
> Thank you for your detailed review comments. Here are our responses:
>
> > • The LoRA-Merge section in **Figure 2 is still not clear**. I suggest that the authors enhance the caption to describe the meaning of each block in the figure.
> >
>
> Regarding the clarity of Figure 2, we fully agree with your suggestion. We have enhanced the caption by adding detailed descriptions of each block in the "LoRA-Merge" section and included a reference to the appendix in the main text to help readers better understand the implementation details.
>
> > • The paper has undergone significant modifications. I recommend that the authors **highlight the changes in a different color** to better showcase these updates or provide a detailed explanation of the revisions in the general response.
> >
>
> Concerning the visibility of our modifications, we have followed your constructive suggestion and highlighted all changes in blue to better showcase our updates.
>
> > • I am still concerned about the novelty: Regarding the significance of maxq, Appendix D provides experiments for the 4-bit case. **Are there any analyses for different bit-widths?** Additionally, could the distribution of LoRA magnitude also be influenced by initialization or learning rate? Please provide some analysis to further clarify the importance of QBAS.
> >
>
> We appreciate the reviewer’s concern about the novelty and the significance of QBAS across different bit-widths. To address this, we have added experimental results for the 3-bit and 2-bit cases, providing a more comprehensive analysis of QBAS's effectiveness across various bit-widths.
>
> Regarding the potential influence of initialization and learning rate on the distribution of LoRA magnitude, we have conducted a detailed analysis. Among various factors, we identified that adjusting the learning rate for different bit-widths is theoretically the closest method to QBAS. This relationship is particularly apparent under the SGD optimizer, where the two approaches are theoretically equivalent, but differences emerge under the Adam optimizer:
>
> ---
>
> #### **Analysis: SGD vs. Adam**
>
> SGD directly updates the parameter $\theta_t$ using the gradient $\nabla_{\theta} J(\theta_t)$ scaled by the learning rate $\eta$:
>
> $$
> \theta_{t+1} = \theta_t - \eta \cdot \nabla_{\theta} J(\theta_t)
> $$
>
> In contrast, Adam dynamically adjusts parameter updates by combining momentum ($m_t$) and adaptive scaling ($v_t$):
>
> $$
> \theta_{t+1} = \theta_t - \eta \cdot \frac{\hat{m}_t}{\sqrt{\hat{v}_t} + \epsilon}
> $$
>
> With:
> $$
> m_t = \beta_1 \cdot m_{t-1} + (1 - \beta_1) \cdot g_t, \quad
> v_t = \beta_2 \cdot v_{t-1} + (1 - \beta_2) \cdot g_t^2
> $$
> $$
> \hat{m}_t = \frac{m_t}{1 - \beta_1^t}, \quad \hat{v}_t = \frac{v_t}{1 - \beta_2^t}
> $$
>
> #### **Difference Between Dividing by `maxq` and Adjusting the Learning Rate to `1/maxq`**
>
> #### **In SGD**
> Case 1: Divide gradient by `maxq`
>    $$
>    \theta_{t+1} = \theta_t - \eta \cdot \frac{\nabla_{\theta} J(\theta_t)}{\text{maxq}}
>    $$
>
> Case 2: Adjust the learning rate to  `1/maxq`
>    $$
>    \theta_{t+1} = \theta_t - \frac{\eta}{\text{maxq}} \cdot \nabla_{\theta} J(\theta_t)
>    $$
>
> For **SGD**, these two approaches are mathematically equivalent.
>
> #### **In Adam**
>  Case 1: Divide gradient by  `1/maxq`
>    The calculations for both the first and second moments are affected:
>    $$
>    m_t = \beta_1 \cdot m_{t-1} + (1 - \beta_1) \cdot \frac{g_t}{\text{maxq}}
>    $$
>    $$
>    v_t = \beta_2 \cdot v_{t-1} + (1 - \beta_2) \cdot \left(\frac{g_t}{\text{maxq}}\right)^2
>    $$
>
> Case 2: Adjust the learning rate to  `1/maxq`
>    $$
>    \theta_{t+1} = \theta_t - \frac{\eta}{\text{maxq}} \cdot \frac{\hat{m}_t}{\sqrt{\hat{v}_t} + \epsilon}
>    $$
>
> For **Adam**, these two approaches are not equivalent due to the influence of `maxq` on the computation of $m_t$ and $v_t$. Dividing the gradient by `maxq` alters both moment estimates, whereas adjusting the learning rate does not.
>
> ---
>
> ### **Further Experiments**
>
> To further explore this, we conducted additional experiments on Flan v2 LLaMA 7B-4bit using a constant learning rate. The results demonstrate that QBAS outperforms direct learning rate adjustments . These findings emphasize QBAS’s superior performance,
>
> | Learning Rate | QBAS | humanities | STEM | social sciences | other | AVG |
> | --- | --- | --- | --- | --- | --- | --- |
> | 1.33e-5 | ❌ | 43.1 | 35.9 | 53.2 | 52.4 | 45.9 |
> | 2e-4 | ✅ | 43.4 | 37.5 | 56.5 | 53.7 | **47.4** |

---

> > ### Comment · Reviewer_2qNk · 2024-11-25
> >
> > Dear Authors,
> >
> > Thank you for your thoughtful and detailed rebuttal. Most of my concerns have been addressed. I therefore have raised my score from 3 to 5.
> >
> > However, as noted by reviewer FBJK, the method shows limited performance on datasets beyond MMLU. While the paper provides valuable insights, it would benefit from including additional datasets and models to further strengthen its contribution and demonstrate its broader applicability.
> >
> > Best regards

---

### Official Review · Reviewer_jKth · 2024-10-29

**Soundness:** 3
**Presentation:** 2
**Contribution:** 3
**Rating:** 5
**Confidence:** 4

**Summary:**

This paper proposes a novel quantization-aware, parameter-efficient tuning method for large language models (LLMs) based on QA-LoRA. The approach reparameterizes both scaling and zero factors to mitigate quantization errors. Experimental results demonstrate its effectiveness.

**Strengths:**

1. Deployment is a crucial consideration for large language models (LLMs), with quantization-aware training being one of the key challenges in their deployment.

2. The proposed method is simple yet promising, as indicated by the experimental results.

**Weaknesses:**

1. While I understand the core idea the author aims to convey, I believe the writing could be improved by providing a more general analysis of approaches to quantization-aware (QA) training or reparameterization of quantization parameters. This would be more important to the community and the contributions will be more important.
2. The contributions appear to be incremental.
3. Figure 2 could be enhanced: it currently labels the LoRA parameters 𝐴 and 𝐵 for QLoRA, but lacks annotations for QA-LoRA and the proposed method. Additionally, the concept of Holistic LoRA is not clearly conveyed.
4. This method seems have limitations which only support group-wise quantization.

**Questions:**

1. Any inference time results?
2. other questions please refer to the weaknesses.

---

> ### Author Response · Authors · 2024-11-20
>
> > While I understand the core idea the author aims to convey, I believe the writing could be improved by providing a more general analysis of approaches to quantization-aware (QA) training or reparameterization of quantization parameters. This would be more important to the community and the contributions will be more important.
> >
>
> Thank you for your insightful suggestion about providing a more general perspective. While we appreciate the value of a broader analytical framework, our current focus on the intersection of LoRA and quantization is deliberate. In our Related Work section, we already cover Fine-Tuning of Quantized Parameters, which represents the main developments in QA training. We chose to emphasize the LoRA+quantization approach specifically because it helps readers intuitively understand the efficiency of our method. While a more general analysis could be valuable, we believe our current presentation effectively communicates our technical contributions while maintaining accessibility for readers.
>
> > The contributions appear to be incremental.
> >
>
> We respectfully disagree with the assessment of LoQA as incremental. LoQA introduces two fundamental innovations that advance the field of LLM quantization: pioneering the concept of merging LoRA into scale quantization parameters and developing QBAS to address abnormal magnitude issues in scale parameter learning. These advances are particularly significant for practical deployment, as most widely-used PTQ methods (such as Omniquant, AffineQuant, GPTQ, QuaRot) compress zero points to integers or eliminate them entirely (like Smooth Quant, AWQ) for acceleration. Additionally, many acceleration libraries like Marlin do not support floating-point zero points, making our scale parameter innovation especially valuable.
>
> The effectiveness of our approach is clearly demonstrated in our ablation studies (Section 4.2), where learning scale parameters achieves significantly better accuracy than increasing LoRA rank, despite using only half the number of parameters.
>
> > Figure 2 could be enhanced: it currently labels the LoRA parameters 𝐴 and 𝐵 for QLoRA, but lacks annotations for QA-LoRA and the proposed method. Additionally, the concept of Holistic LoRA is not clearly conveyed.
> >
>
> Thank you for this observation. We have revised Figure 2 to provide a more comprehensive and clear visualization of all methods. The updated figure now includes complete parameter annotations for QLoRA, QA-LoRA, and our proposed method. We have explicitly labeled all LoRA parameters (A, B, A', B') across different approaches and added clear illustrations of the Holistic LoRA concept, showing how the two LoRA variants interact with scale and zero point components. These improvements should make the architectural differences and innovations much clearer to readers.
>
> > This method seems have limitations which only support group-wise quantization.
> >
>
> While our method primarily focuses on group-wise quantization, this design choice offers significant advantages. Group-wise quantization provides excellent inference acceleration for large models while minimizing accuracy loss at low bit-widths, making it widely adopted in practice. Most state-of-the-art uniform quantization PTQ methods use group-wise quantization as their primary benchmark, and our method naturally adapts to this paradigm.
>
> > Any inference time results?
> >
>
> LoQA's key advantage lies in its compatibility with established weight-only quantization acceleration frameworks. We have implemented benchmarks using both MARLIN and BitBLAS on an A100 GPU (batch size=16), demonstrating significant speedup ratios. The detailed computational costs and acceleration metrics have been incorporated into the paper, showing how LoQA effectively leverages existing optimization techniques for efficient inference.
>
> | Model | TFLOP/s | Speedup |
> | --- | --- | --- |
> | Llama7B | 63.788 | 2.71 |
> | Llama13B | 76.907 | 3.31 |
> | Llama33B | 87.907 | 3.5 |
> | Llama65B | 92.807 | 3.61 |
> | Falcon180B | 104.5 | 3.81 |

---

> > ### Comment · Area_Chair_Vy59 · 2024-11-25
> >
> > Dear reviewer Jkth,
> >
> > As the deadline for discussion is ending soon. Please respond to the authors to indicate you have read their rebuttal. If you have more questions, now is the time to ask.
> >
> > AC

---

> ### Comment · Reviewer_jKth · 2024-11-27
>
> Thanks for the authors' efforts and responses. I still have some concerns about the contributions of this paper. I am not convinced by the response of the "incremental" weakness.  I choose to maintain my score.

---

### Official Review · Reviewer_n8i8 · 2024-10-31

**Soundness:** 2
**Presentation:** 3
**Contribution:** 2
**Rating:** 5
**Confidence:** 4

**Summary:**

This paper aims to enable efficient LLM fine-tuning by combining quantization with LoRA. Rather than keeping LoRA inference in FP/BF16 during the forward pass, it proposes a holistic quantized LoRA approach where LoRA is fused into the quantization set to enable fully quantized inference. Additionally, the paper introduces a method to determine the LoRA scaling factor based on the target bit precision.

The proposed method achieves higher fine-tuning performance and training efficiency compared to QLoRA and QA-LoRA at 4/3/2-bit precision.

**Strengths:**

- The related work and motivation are clearly presented, making the problem the paper addresses easy to understand.
- The holistic quantized LoRA approach follows a similar direction to QA-LoRA, aiming to achieve fully quantized inference by fusing LoRA. The paper introduces a way to learn both the scaling and zero-point parameters, which adds flexibility to the method.

**Weaknesses:**

- The holistic quantization method formulation is somewhat difficult to follow. Figure 2 is overly simplified, and would benefit from revision to visually clarify the equations in Section 3.2.
- Experiments are limited to the LLaMA family, and there is a lack of results on LLaMA-3. If dataset is an issue, it may be preferable to use more recent open-source datasets (e.g., SlimPajama) instead of the relatively old and small sample Alpaca dataset. Alternatively, comparing performance on domain-specific fine-tuning tasks (e.g., GSM8K, OASST1) as done in LoftQ/LQ-LoRA could provide additional insights.
- The main contribution of this paper appears to be preserving the quantization set in inference. However, the proposed method lacks significant innovation compared to QA-LoRA, and there is no report on the actual efficiency gains achieved during fully quantized inference. Section 4.3 only briefly mentions the potential for kernel optimizations, which seems insufficient given that fullly quantized inference is the main focus of this paper.

In summary, while the proposed method consistently outperforms QA-LoRA and addresses an important problem, the limited experimental results and unclear explanations of key methods indicate room for improvement.

**Questions:**

- In Section 3.2, the proposed method is still not entirely clear. On Line 249, what exactly do you mean by “two LoRA variants”?

---

> ### Author Response · Authors · 2024-11-20
>
> > The holistic quantization method formulation is somewhat difficult to follow. Figure 2 is overly simplified, and would benefit from revision to visually clarify the equations in Section 3.2.
> >
>
> Thank you for this valuable feedback regarding the clarity of our presentation. We have significantly enhanced Figure 2 to provide a more comprehensive visual explanation of our method. The key improvements include:
>
> 1. Illustrating HQ-LoRA using two distinct LoRA components for better visualization
> 2. Explicitly showing the mergeable components in the LoRA-Merge section, corresponding to equations (8) and (9) in the original text
> 3. Clearly demonstrating LoQA's novel contributions compared to previous methods
>
> To make the holistic quantization method formulation more accessible, we have also included a special case analysis in Section 3.2 with group size=1. We believe this simplified example significantly aids in understanding the general formulation.
>
> > Experiments are limited to the LLaMA family, and there is a lack of results on LLaMA-3. If dataset is an issue, it may be preferable to use more recent open-source datasets (e.g., SlimPajama) instead of the relatively old and small sample Alpaca dataset. Alternatively, comparing performance on domain-specific fine-tuning tasks (e.g., GSM8K, OASST1) as done in LoftQ/LQ-LoRA could provide additional insights.
> >
>
> Thank you for this detailed feedback. We have expanded our experiments and analysis in several important ways:
>
> 1. Model Generalization:
> We have extended our evaluation beyond the LLaMA family by testing on OPT-6.7B with Flan-v2:
>
>
>     | Method | Dataset | #Bit | mmlu(5-shot) |
>     | --- | --- | --- | --- |
>     | OPT 6.7B | - | 16 | 24.57 |
>     | QA-LoRA | FLAN-v2 | 4 | 29.85 |
>     | LoQA | FLAN-v2 | 4 | 33.78 |
> 2. Domain-Specific Tasks:
>
>     We've added experiments on LLaMA3 using GPTQ quantization for GSM8K:
>
>     | Method | Dataset | #Bit | GSM8K(0-shot) |
>     | --- | --- | --- | --- |
>     | LLaMA3-8B-GPTQ | - | 4 | 5.99% |
>     | QA-LoRA | GSM8K | 4 | 52.62% |
>     | LoQA | GSM8K | 4 | 56.41% |
>
>
> > The main contribution of this paper appears to be preserving the quantization set in inference. However, the proposed method lacks significant innovation compared to QA-LoRA, and there is no report on the actual efficiency gains achieved during fully quantized inference. Section 4.3 only briefly mentions the potential for kernel optimizations, which seems insufficient given that fullly quantized inference is the main focus of this paper.
> >
>
> Regarding inference speed, we want to clarify that LoQA's primary advantage isn't direct acceleration compared to QA-LoRA, as both methods aim to integrate LoRA into low-bit models. However, LoQA is compatible with existing weight-only quantization acceleration frameworks like MARLIN[1] and BitBLAS[2]. We have conducted acceleration tests on an A100 GPU with batch size 16, and the detailed computational costs and speedup ratios have been added to the paper.
>
> | Model | TFLOP/s | Speedup |
> | --- | --- | --- |
> | Llama7B | 63.788 | 2.71 |
> | Llama13B | 76.907 | 3.31 |
> | Llama33B | 87.907 | 3.5 |
> | Llama65B | 92.807 | 3.61 |
> | Falcon180B | 104.5 | 3.81 |
>
> [1]MARLIN: Mixed-Precision Auto-Regressive Parallel Inference on Large Language Models
>
> [2]Ladder: Enabling Efficient Low-Precision Deep Learning Computing through Hardware-aware Tensor Transformation
>
> > In Section 3.2, the proposed method is still not entirely clear. On Line 249, what exactly do you mean by “two LoRA variants”?
> >
>
> In Section 3.2, "two LoRA variants" refers to:
>
> 1. The A'B' matrices that fine-tune the scale component
> 2. The AB matrices that fine-tune the zero point component
>
> We have visually illustrated this dual-LoRA structure in Figure 2 to aid understanding. The diagram shows how these two distinct LoRA components work together.

---

> > ### Comment · Reviewer_n8i8 · 2024-11-22
> >
> > Thank you for the detailed response. The experimental results you presented partially addressed my concerns. However, I still find that the contribution of LoQA does not significantly deviate from that of the existing QA-LoRA, which remains a concern. Additionally, I noticed that the accuracy drop for LLaMA-3-8B-GPTQ 4-bit is excessively large, which raises some questions about the experimental setup. Therefore, I will maintain my score.

---

> > > ### Author Response · Authors · 2024-11-23
> > >
> > > Dear Reviewer,
> > >
> > > Thank you for your feedback. Here are our responses:
> > >
> > > > Thank you for the detailed response. The experimental results you presented partially addressed my concerns. However, I still find that the contribution of LoQA does not significantly deviate from that of the existing QA-LoRA, which remains a concern.
> > > >
> > >
> > > As we discussed with other reviewers, we respectfully disagree with the characterization of LoQA as merely incremental. Like QA-LoRA, our method LoQA employs LoRA and supports reparameterization of LoRA into quantized weights. However, LoQA makes two fundamental contributions beyond QA-LoRA: (1) it pioneers the novel concept of merging LoRA into scale quantization parameters, and (2) introduces QBAS to address the abnormal magnitude issues in learning scale parameters. These advances represent significant improvements in both versatility and accuracy, supported by the following evidence:
> > >
> > > 1. Firstly, LLM inference is primarily memory-bounded, where weight-only quantization methods accelerate computation by reducing memory access. Most widely-used PTQ methods compress zero points to integers (e.g., Omniquant[1], AffineQuant[2], FlatQuant[3]) or eliminate them entirely (e.g., Smooth Quant[4], AWQ[5]) for further acceleration. Additionally, some acceleration libraries like Marlin do not support floating-point zero points. Therefore, our innovation of introducing LoRA to scale parameters has substantial practical implications for deploying these methods.
> > >
> > >     [1] Shao W, Chen M, Zhang Z, et al. Omniquant: Omnidirectionally calibrated quantization for large language models[J]. arXiv preprint arXiv:2308.13137, 2023.
> > >
> > >     [2]Ma Y, Li H, Zheng X, et al. Affinequant: Affine transformation quantization for large language models[J]. arXiv preprint arXiv:2403.12544, 2024.
> > >
> > >     [3]Sun Y, Liu R, Bai H, et al. FlatQuant: Flatness Matters for LLM Quantization[J]. arXiv preprint arXiv:2410.09426, 2024.
> > >
> > >     [4]Xiao G, Lin J, Seznec M, et al. Smoothquant: Accurate and efficient post-training quantization for large language models[C]//International Conference on Machine Learning. PMLR, 2023: 38087-38099.
> > >
> > >     [5]Lin J, Tang J, Tang H, et al. AWQ: Activation-aware Weight Quantization for On-Device LLM Compression and Acceleration[J]. Proceedings of Machine Learning and Systems, 2024, 6: 87-100.
> > >
> > > 2. Next, as demonstrated in our ablation studies (Section 4.2), when both zero points and scales are learnable, learning scale parameters achieves significantly better accuracy than increasing LoRA rank, despite using only half the number of parameters. This substantial improvement in accuracy with fewer parameters clearly demonstrates the effectiveness of our approach to scale parameter learning. Extensive experiments demonstrate that LoQA consistently outperforms previous fine-tuning methods that maintain quantized formats, and in many cases, matches the performance of state-of-the-art 4+16 bit methods. Notably, in ultra-low bit-width scenarios, LoQA's effectiveness is even more pronounced, with its 2-bit version surpassing the current 2+16-bit state-of-the-art method by 4.7% and even outperforming the original 16-bit model.
> > >
> > > We have also added additional analyses of QBAS in the revised manuscript, further demonstrating its effectiveness. We encourage you to review these updates and would be happy to discuss them further.
> > >
> > > > Additionally, I noticed that the accuracy drop for LLaMA-3-8B-GPTQ 4-bit is excessively large, which raises some questions about the experimental setup. Therefore, I will maintain my score.
> > > >
> > >
> > > Regarding the reported performance drop for LLaMA-3-8B-GPTQ 4-bit, we acknowledge the concern and have investigated this issue further. The training and testing scripts for GSM8K primarily follow the settings from LoftQ [1]. A potential reason for the poor performance is the use of 0-shot testing with specific formatting. To provide additional context, we have included the results of the 16-bit baseline models as a reference:
> > >
> > > | Method | Dataset | #Bit | GSM8K(0-shot) |
> > > | --- | --- | --- | --- |
> > > | Meta-Llama-3-8B | - | 16 | 5.00% |
> > > | Meta-Llama-3-8B-Instruct | - | 16 | 37.68% |
> > >
> > > Additionally, we have updated our supplementary materials to include the exact training and testing scripts used for this experiment. If you have further questions or suggestions about the experimental setup, we would be more than happy to discuss them in detail.
> > >
> > > [1]Li Y, Yu Y, Liang C, et al. Loftq: Lora-fine-tuning-aware quantization for large language models[J]. arXiv preprint arXiv:2310.08659, 2023.

---

### Official Review · Reviewer_45yq · 2024-11-03

**Soundness:** 2
**Presentation:** 3
**Contribution:** 2
**Rating:** 5
**Confidence:** 3

**Summary:**

The paper introduces Low-Rank Quantization Adaptation, called LoQA, a new approach for fine-tuning large language models (LLMs) while preserving quantization. LoQA consists of two techniques: (1) Holistic Quantization Low-Rank Adaptation (HQ-LoRA): This approach fine-tunes all quantized parameters, enabling broader optimization without losing the quantized model structure. (2) Quantized Bit-Aware Scaling (QBAS): This technique dynamically adjusts scaling factors based on bit-widths, enhancing performance stability across various quantization levels. LoQA is tested against state-of-the-art quantization methods and is effective across different model sizes and bit configurations. It yields significant improvements, particularly under ultra-low bit-width scenarios.

**Strengths:**

+ LoQA introduces a novel quantization-aware fine-tuning method that can support quantized LLMs to effectively balance model compression with performance.
+ Experiments show that LoQA consistently outperforms previous methods in accuracy and efficiency, even in ultra-low bit-width configurations.
+ The method supports multiple bit-widths and integrates smoothly with existing post-training quantization methods, making it adaptable to a wide range of applications.
+ LoQA is evaluated using the MMLU and common sense QA tasks, with consistent results.

**Weaknesses:**

- LoQA requires more training time and memory compared to some baseline methods, which may hinder its practicality in resource-constrained environments (e.g., compared with QA-LoRA)
- The study did not use the latest datasets or the newest training paradigms, which might limit the generalizability of the results to other, more current LLMs (as also mentioned in Appendix).

**Questions:**

Would you please describe how LoQA could be implemented in production?

---

> ### Author Response · Authors · 2024-11-20
>
> Thank you for your detailed review. We would like to address each of your points:
>
> > LoQA requires more training time and memory compared to some baseline methods, which may hinder its practicality in resource-constrained environments.
> >
>
> Regarding training costs, we provide a comprehensive analysis in Section 4.3 (Training Cost), which demonstrates that LoQA requires approximately only 1.3 times the training time of QA-LoRA, while LoQA-S demands even less than this 1.3-fold increase. For context, according to the QA-LoRA study, QLoRA necessitates approximately twice the training time of QA-LoRA [1]. These results indicate that, under equivalent optimization conditions, LoQA achieves optimal results with a balanced training cost.
> [1]Xu Y, Xie L, Gu X, et al. Qa-lora: Quantization-aware low-rank adaptation of large language models[J]. arXiv preprint arXiv:2309.14717, 2023.
>
> > The study did not use the latest datasets or the newest training paradigms, which might limit the generalizability of the results to other, more current LLMs.
> >
>
> Concerning the use of datasets and training paradigms, we acknowledge this limitation in Appendix A. While using newer training paradigms or advanced PTQ/LoRA methods could potentially improve results, our current experimental setup effectively demonstrates our method's core advantages. A complete reproduction across all newer models and methods would require substantial computational resources without necessarily providing additional insights into our method's fundamental contributions.
>
> > Implementing LoQA in production may involve significant effort due to the complexity of fine-tuning across various bit-widths, which could add deployment challenges.
> >
>
> We respectfully disagree with the concern about deployment challenges. LoQA does not introduce significant overhead for fine-tuning across different bit-widths. In fact, our method is more deployment-friendly than previous approaches as it doesn't rely on floating-point zero points, making it widely compatible with various PTQ methods and acceleration kernels. We have illustrated this clearly in Figure 2 and elaborated in our response to reviewer FBJK.
>
> > Would you please describe how LoQA could be implemented in production?
> >
>
> As for production implementation, LoQA is particularly well-suited for resource-constrained environments. It enables efficient fine-tuning while maintaining quantization properties, making it ideal for edge devices or scenarios with limited resources. For instance, models deployed locally can be fine-tuned on user-specific data with limited memory while directly producing quantized models for serving. The method's compatibility with various PTQ methods and acceleration kernels, due to its independence from floating-point zero points, ensures broad applicability in production settings.

---

> > ### Comment · Area_Chair_Vy59 · 2024-11-25
> >
> > Dear reviewer 45yq,
> >
> > As the deadline for discussion is ending soon. Please respond to the authors to indicate you have read their rebuttal. If you have more questions, now is the time to ask.
> >
> > AC

---

### Official Review · Reviewer_FBJK · 2024-11-03

**Soundness:** 3
**Presentation:** 2
**Contribution:** 2
**Rating:** 5
**Confidence:** 3

**Summary:**

This paper extends the QA-LoRA method by introducing Low-Rank Quantization Adaptation (LoQA) to enhance the fine-tuning of large language models (LLMs) within a quantized framework. LoQA addresses the integration challenges between quantization and Low-Rank Adaptation (LoRA), proposing Holistic Quantization Low-Rank Adaptation (HQ-LoRA), a quantization-compatible adaptation technique that allows finetuning of all quantization parameters (including scale and zero points) resulting in improved model performance. Additionally, the paper introduces Quantized Bit-aware scaling (QBAS), which adjusts LoRA scaling to account for the impact of integer weights at varying bit-widths. Overall, LoQA extends the QA-LoRA paper by modifying not only the quantization zero points (as in QA-LoRA) but also the quantization scale.

**Strengths:**

Strengths:

-Considering the quantization scaling as well during finetuning gives much more representation power compared to earlier approaches like QA-LoRA, which focus only on quantization zero points. Also, as demonstrated by empirical results, it usually results in better generalization across diverse set of models and bit budgets.

-LoQA especially outperforms previous methods on low-bit budgets, which would make it useful for ultra resource-efficient settings.

-In the 4bit regime, LoQA seems to perform competitively to algorithms like IR-QLoRA that use a mix of 4/16 in their inference deployment

-Given that it doesn’t carry 16bit LoRA parameters to the inference phase (as opposed to methods like QLoRA), it would bring considerable inference efficiency compared to "4+16" type of methods.

**Weaknesses:**

In general, this method is interesting - however, in terms of novelty,  this work is indeed an incremental work on top of the QA-LoRA work. Moreover, this paper could benefit from improvements in the presentation and description of the experiments.

Weaknesses:

-The paper presentation could be improved, especially in the experiment section. Implementation details are underspecified in many parts. Given that this research effort is based on empirical results, this lack of experimental setting would make it hard to reproduce the results for the community.

-For experiments comparing LoQA with QA-LoRA (as well as other similar methods), the corresponding ranks of the methods are not reported in most cases, which makes the reader question if the improvement could be due to using more parameters by LoQA or not.

-There’s lot of focus on MMLU evaluation after finetuning on alpaca/Flan datasets, which might not be representative of how this algorithm would actually work for other type of tasks/datasets. The only exception, Table 5, focuses on commonsense reasoning, which the performance is very close to QA-LoRA, while it uses half of your training complexity by having a single set of LoRA parameters for zero points.

-Table captions in many cases are not "self contained" and often have a very general caption, which make it hard to go through different tables and understand the results.

**Questions:**

-Regarding QBAS: the scaling that you proposed is then being multiplied by a hyperparameter. Are you using the same $alpha$ across different bits?

---

> ### Author Response · Authors · 2024-11-20
>
> Thank you for your thorough and constructive feedback. Your comments have helped us significantly improve the clarity and rigor of our manuscript. We have carefully addressed each point to enhance reproducibility, clarify our contributions, and better communicate our experimental results. We believe these revisions have strengthened the paper and made our technical contributions more accessible to the research community.
>
> > In general, this method is interesting - however, in terms of novelty, this work is indeed an incremental work on top of the QA-LoRA work.
> >
>
> We respectfully disagree with the characterization of LoQA as merely incremental. Like QA-LoRA, our method LoQA employs LoRA and supports reparameterization of LoRA into quantized weights. However, LoQA makes two fundamental contributions beyond QA-LoRA: (1) it pioneers the novel concept of merging LoRA into scale quantization parameters, and (2) introduces QBAS to address the abnormal magnitude issues in learning scale parameters. These advances represent significant improvements in both versatility and accuracy, supported by the following evidence:
>
> 1. Firstly, LLM inference is primarily memory-bounded, where weight-only quantization methods accelerate computation by reducing memory access. Most widely-used PTQ methods compress zero points to integers (e.g., Omniquant[1], AffineQuant[2], FlatQuant[3]) or eliminate them entirely (e.g., Smooth Quant[4], AWQ[5]) for further acceleration. Additionally, some acceleration libraries like Marlin do not support floating-point zero points. Therefore, our innovation of introducing LoRA to scale parameters has substantial practical implications for deploying these methods.
>
>     [1] Shao W, Chen M, Zhang Z, et al. Omniquant: Omnidirectionally calibrated quantization for large language models[J]. arXiv preprint arXiv:2308.13137, 2023.
>
>     [2]Ma Y, Li H, Zheng X, et al. Affinequant: Affine transformation quantization for large language models[J]. arXiv preprint arXiv:2403.12544, 2024.
>
>     [3]Sun Y, Liu R, Bai H, et al. FlatQuant: Flatness Matters for LLM Quantization[J]. arXiv preprint arXiv:2410.09426, 2024.
>
>     [4]Xiao G, Lin J, Seznec M, et al. Smoothquant: Accurate and efficient post-training quantization for large language models[C]//International Conference on Machine Learning. PMLR, 2023: 38087-38099.
>
>     [5]Lin J, Tang J, Tang H, et al. AWQ: Activation-aware Weight Quantization for On-Device LLM Compression and Acceleration[J]. Proceedings of Machine Learning and Systems, 2024, 6: 87-100.
>
> 2. Next, as demonstrated in our ablation studies (Section 4.2), when both zero points and scales are learnable, learning scale parameters achieves significantly better accuracy than increasing LoRA rank, despite using only half the number of parameters. This substantial improvement in accuracy with fewer parameters clearly demonstrates the effectiveness of our approach to scale parameter learning. Extensive experiments demonstrate that LoQA consistently outperforms previous fine-tuning methods that maintain quantized formats, and in many cases, matches the performance of state-of-the-art 4+16 bit methods. Notably, in ultra-low bit-width scenarios, LoQA's effectiveness is even more pronounced, with its 2-bit version surpassing the current 2+16-bit state-of-the-art method by 4.7\% and even outperforming the original 16-bit model.

---

> ### Author Response · Authors · 2024-11-20
>
> > -The paper presentation could be improved, especially in the experiment section. Implementation details are underspecified in many parts. Given that this research effort is based on empirical results, this lack of experimental setting would make it hard to reproduce the results for the community.
> >
>
> We appreciate this concern about reproducibility. We would like to point out that comprehensive implementation details are provided in Appendix B under the SETTINGS section, where we document most critical experimental parameters. To further enhance reproducibility, we have added Table 9 which explicitly lists key training hyperparameters. Additionally, our released code repository includes complete reproduction procedures and detailed implementation specifics.
>
> > -For experiments comparing LoQA with QA-LoRA (as well as other similar methods), the corresponding ranks of the methods are not reported in most cases, which makes the reader question if the improvement could be due to using more parameters by LoQA or not.
> >
>
> We thank the reviewer for this important question about parameter efficiency. We would like to clarify that all comparative experiments with previous methods (QA-LoRA, Q-LoRA, and IR-QLoRA) consistently use rank=64. We have now made this explicit in the experimental section for better clarity.
>
> Furthermore, our ablation analysis in Section 4.2 (Table 8) specifically investigates the impact of parameter count. The results demonstrate that the performance improvements of LoQA are not simply due to increased parameter count. While increasing parameters through higher rank yields only marginal improvements, our method achieves significant performance gains through more effective parameter utilization.
>
> > -There’s lot of focus on MMLU evaluation after finetuning on alpaca/Flan datasets, which might not be representative of how this algorithm would actually work for other type of tasks/datasets. The only exception, Table 5, focuses on commonsense reasoning, which the performance is very close to QA-LoRA, while it uses half of your training complexity by having a single set of LoRA parameters for zero points.
> >
>
> We appreciate this concern regarding evaluation diversity. Our choice to evaluate on MMLU after Alpaca/Flan fine-tuning aligns with previous works (QA-LoRA, IR-QLoRA) as this setup effectively assesses both instruction-following capabilities and knowledge retention.
>
> Regarding the commonsense reasoning results, we would like to emphasize that improving upon QA-LoRA's strong baseline of 65.0 on commonsense QA (achieved after Flan-v2 fine-tuning) is particularly challenging. In this context, LoQA's improvement to 65.6 represents a meaningful advance. To further validate our method's effectiveness across different scenarios, we have expanded Table 5 to include additional comparison results between LoQA and QA-LoRA on the Alpaca dataset, which further demonstrates the consistent advantages of our approach.
>
> > -Table captions in many cases are not "self contained" and often have a very general caption, which make it hard to go through different tables and understand the results.
> >
>
> We thank the reviewer for this feedback regarding table clarity. We have revised all table captions to be more comprehensive and self-contained.
>
> > Regarding QBAS: the scaling that you proposed is then being multiplied by a hyperparameter. Are you using the same alpha across different bits?
> >
>
> Yes, we maintain consistent hyperparameters across different bit-widths, including the same alpha value, except for those parameters explicitly specified in the tables. This consistency ensures fair comparisons and reliable experimental results.

---

> > ### Comment · Area_Chair_Vy59 · 2024-11-25
> >
> > Dear reviewer FBJK,
> >
> > As the deadline for discussion is ending soon. Please respond to the authors to indicate you have read their rebuttal. If you have more questions, now is the time to ask.
> >
> > AC

---

> > ### Comment · Reviewer_FBJK · 2024-11-25
> >
> > Thank you for your response, as well as the improvements in the paper presentation.
> > While I appreciate the clarifications you've provided, I remain unconvinced that the novelty of your approach significantly advances beyond the existing QA-LoRA method. The proposed changes still seem to represent a relatively marginal improvement over QA-LoRA, both in terms of methodology and results. Additionally, the gains observed on non-MMLU benchmarks appear to be quite small compared to QA-LoRA, which raises concerns about the overall impact of the approach in the community. Therefore, I would keep my current score.

---

### Author Response · Authors · 2024-11-25

We sincerely thank all the reviewers and area chairs for their time and efforts in reviewing our paper. We appreciate the insightful and constructive comments provided by the reviewers. We have addressed each of the points raised and made revisions accordingly. Below is a summary of the main contributions:

**Novelty**

We propose a novel and efficient fine-tuning method, LoQA, for quantized large language models. This method seamlessly integrates Low-Rank Adaptation (LoRA) with group-wise quantization into a unified framework. Compared to previous work, LoQA makes two fundamental contributions: (1) it introduces the novel concept of merging LoRA with scale quantization parameters, and (2) it introduces QBAS to address the abnormal magnitude issues in learning scale parameters. These advances represent significant improvements in the versatility and accuracy of such methods.

**Strong and Comprehensive Empirical Results**

The experimental results demonstrate consistent improvements across different LLM architectures and quantization bit-widths. Our experiments cover a variety of architectures (e.g., LLaMA1, LLaMA2, LLaMA3, OPT) and model sizes, and showcase results across different datasets (Alpaca, Flan v2, GSM8K) and bit-widths. LoQA exhibits its adaptability across models and sizes, with significant improvements in both Commonsense QA and MMLU accuracy. Despite using only half the number of parameters, LoQA achieves a substantial accuracy boost. Extensive experiments show that LoQA consistently outperforms previous fine-tuning methods that retain quantized formats and, in many cases, matches the performance of state-of-the-art 4+16 bit methods. Notably, in ultra-low bit-width scenarios, LoQA's effectiveness is even more pronounced, with its 2-bit version outperforming the current 2+16 bit state-of-the-art method by 4.7%, and even surpassing the original 16-bit model.

**Inference Efficiency**

LoQA preserves the inference speed advantages of quantization by merging the low-rank adaptation weights, while also being compatible with the latest inference engines such as Marlin and BitBlas. This results in significant improvements in inference speed, which is crucial for deploying LLMs in resource-constrained environments.

We also appreciate the constructive suggestions and questions raised by the reviewers, which have led to fruitful discussions and additional experiments. We have made appropriate revisions based on these suggestions, with all changes marked in blue for easy reference. We believe these revisions further strengthen the quality of the paper. Below are the major revisions:

1.We have re-drawn Figure 2 to improve clarity, and added additional explanations for the "LoRA-Merge" section with different color blocks to aid reader comprehension.
2.We revised the titles of each table to make them more self-contained and reader-friendly, and added explanations for parameters like “rank” and “4+16-bit” to prevent any misunderstandings.
3.We added comprehensive implementation details in Appendix A.
4.We included a comparison of Commonsense QA results on the Alpaca dataset to further highlight the accuracy advantages of our method.
5.We included evaluation results using domain-specific datasets GSM8K on LLaMA3.
6.We added Appendix D, which provides a visual analysis of the QBAS method, comparing magnitude changes before and after using QBAS to clearly demonstrate its effect.
6.We included Appendix F, where we tested our method on Flan-v2 with OPT-6.7B, expanding our evaluation beyond the LLaMA series.
7.We added Appendices G and F, which include detailed information on training and inference speed experiments.

Finally, **we have addressed all raised questions and concerns**, and detailed responses to individual reviewers are provided below. We look forward to any additional comments.

---

### Meta-Review · Area_Chair_Vy59 · 2024-12-18

**Metareview:**

This paper proposes Low-Rank Quantization Adaptation (LoQA), an approach to fine-tuning LLMs within a quantized framework that introduces two primary techniques: Holistic Quantization LoRA (HQ-LoRA) and Quantized Bit-Aware Scaling (QBAS), which aim to enhance model performance and efficiency during quantization. The research demonstrates consistent improvements, particularly in ultra-low bit-width scenarios, with most reviewers acknowledging its potential to balance model compression and performance (45yq, jKth, 2qNk). Reviewers noted strengths including the method's ability to fine-tune quantization parameters comprehensively, outperform previous approaches like QA-LoRA, and show promising results across different model sizes and bit configurations (FBJK, 45yq, n8i8).

However, the paper exhibits several significant weaknesses that led to its negative rating (all reviewers gave a 5). Reviewers consistently pointed out limited novelty, with the work being largely incremental to previous research (2qNk, jKth, FBJK). Experimental limitations were prominent, including underspecified implementation details, restricted dataset usage (primarily Alpaca/Flan), and lack of comprehensive comparisons with other SOTA (FBJK, n8i8, 2qNk). Presentation issues were also critical, with reviewers highlighting unclear figures, inconsistent terminology, missing context in tables, and insufficient explanation of key methodological components (n8i8, 2qNk, jKth). Additionally, concerns were raised about the method's training complexity, limited generalizability, and the absence of clear inference time results (45yq, jKth, 2qNk).

**Additional Comments On Reviewer Discussion:**

All reviewers acknowledged the experiments section. But they are also having the novelty concern, which is the major reason for rejection.

---

### Decision · Program_Chairs · 2025-01-22

Reject